# Diversity of spatiotemporal coding reveals specialized visual processing streams in the mouse cortex

Xu Han [1,2✉], Ben Vermaercke[1,3] & Vincent Bonin [1,2,4,5,6✉]

The cerebral cortex contains diverse neural representations of the visual scene, each enabling distinct visual and spatial abilities. However, the extent to which representations are distributed or segregated across cortical areas remains poorly understood. By determining the spatial and temporal responses of >30,000 layer 2/3 pyramidal neurons, we characterize the functional organization of parallel visual streams across eight areas of the mouse cortex. While dorsal and ventral areas form complementary representations of spatiotemporal frequency, motion speed, and spatial patterns, the anterior and posterior dorsal areas show distinct specializations for fast and slow oriented contrasts. At the cellular level, while diverse spatiotemporal tuning lies along a continuum, oriented and non-oriented spatial patterns are encoded by distinct tuning types. The identified tuning types are present across dorsal and ventral streams. The data underscore the highly specific and highly distributed nature of visual cortical representations, which drives specialization of cortical areas and streams.

[1] Neuro-Electronics Research Flanders, Kapeldreef 75, 3001 Leuven, Belgium. [2] KU Leuven, Department of Biology, 3000 Leuven, Belgium. [3] VIB-KU Leuven Center for Brain & Disease Research, 3000 Leuven, Belgium. [4] VIB, 3001 Leuven, Belgium. [5] imec, 3001 Leuven, Belgium. [6] KU Leuven, Leuven Brain Institute, 3000 Leuven, Belgium. ✉email: hanxu083@gmail.com; vincent.bonin@nerf.be

Visual and sensorimotor processing in mammals depends on a vast sensory cortical processing network whose neurons encode specific aspects of the visual world to enable specific abilities and behavioral goals. At the center of this network are visual cortical areas, which integrate specific visual information from cortical and thalamic pathways[1] to generate visual tuning properties (i.e., selectivity for particular visual features) of increasing complexity. While neurons in the primary visual cortex (V1) are tuned to basic visual features[2,3] (e.g., the orientation of a bar), neurons in higher visual areas (HVAs) are tuned to more complex properties (e.g., pattern motion, the curvature of an object)[4–6].

In primates, dorsal and ventral cortical areas form two broad information streams, which encode specific information and make specific contributions to behavior[7–9]. Involved in motion processing, neurons of the dorsal stream have specific motion tuning properties encoding the location and movement of objects (e.g., pattern motion, direction, and speed selectivity)[10–12]. Involved in scene analysis, neurons of the ventral stream show a variety of spatial tuning properties (color, curvature, etc.) encoding object identity[13–15]. Forming specific connectivity with the magnocellular and parvocellular pathways, the dorsal and ventral streams integrate visual information of distinct spatiotemporal scales and distinct spatial features[16,17]. However, there is considerable overlap between these representations[18], suggesting a more elaborate organization of visual information in the cortex.

In mice, over ten visual cortical areas have been identified, each forming a distinct visual field representation and distinct long-range connectivity[19–21]. Mouse cortical neurons display many of the visual tuning properties observed in cats and primates, including orientation selectivity, tuning for spatial and temporal frequency, motion speed selectivity, and selectivity for stimulus size[22,23]. As in primates, these areas are directly implicated in perception and behavior[24–26]. Accumulating evidence also indicates organized visual pathways[23,27–30] and processing streams[31–37]. In particular, dorsal and ventral HVAs show specific functional tuning biases suggestive of specialized representations and processing streams[29,31–36]. Anterolateral (AL) and posteromedial (PM) areas respectively respond most strongly to fast and slow moving stimuli. Widefield functional imaging of HVAs reported biased responses to stimuli of different spatiotemporal frequencies in dorsal and ventral areas[35] but did not address the diversity and specificity of HVA neural populations. While V1 studies reported nearly exclusively orientation-tuned responses with a diverse orientation selectivity in excitatory neurons[22,38], HVA data indicate differences in orientation selectivity across visual areas with the dorsal areas presenting higher selectivity than V1 and ventral areas[31,32].

However, existing datasets in the mouse focus on a limited subset of visual areas with relatively sparse sampling of the neuron population and restricted sets of stimuli; therefore, the functional relationships amongst populations of visual areas remain poorly characterized. With regard to visual stimuli, most studies in the mouse used drifting gratings as the stimulus of choice for the characterization of visual properties. Some studies used motion stimuli[39–41] but did not characterize detailed spatial and temporal properties. A recent large-scale study[37] recorded responses to a broader set of stimuli but could not characterize differences across visual areas and processing streams.

In this study, we characterized at cellular and mesoscopic levels the organization and specialization of visual processing streams in the mouse cortex. Using rich visual noise stimuli, we probed the multidimensional functional tuning properties of superficial cortical excitatory neurons, addressing selectivity for spatial and temporal frequency, motion speed, and tuning for oriented and non-oriented stimuli. We found diverse and stereotyped visual tuning types with biased distribution across the dorsal and ventral cortical areas. They form the cellular basis of area-specific representations and provide strong functional evidence for parallel specialized processing streams in the mouse visual cortex.

## Results

**Characterization of mouse visual cortical areas with spatiotemporal noise stimuli.** To investigate the organization and specialization of visual processing streams in the mouse cortex, we characterized, using 2-photon calcium imaging[42] in Thy1-GCaMP6s mice[43] (line GP4.12, $N = 10$ mice) and parameterized visual stimuli (Fig. 1a), (azimuth 0 to 100 deg; elevation −30 to 50 deg), the visual receptive field tuning properties of layer 2/3 cortical pyramidal neurons, sampling activity across eight retinotopic visual cortical areas delineated using widefield calcium imaging (Fig. 1b–c, Supplementary Fig. 1, Supplementary Movie 1) including V1 and seven higher visual areas (HVAs). In the primate cortex, ventral and dorsal visual areas show distinct functional properties[44,45]. To investigate whether an analogous organization exists in the mouse, we contrasted tuning properties in V1, LM (lateromedial), dorsal areas AL, RL, AM, and PM (anterolateral, rostrolateral, anteromedial, and posteromedial), and ventral areas LI and POR/P (laterointermediate, postrhinal/posterior areas). Because the borders between POR and P could not always be identified, the results for these two areas are presented as 'POR/P'.

To characterize neurons' tuning properties, we presented spatiotemporal noise stimuli made of random visual patterns of different spatial and temporal frequency (dataset 1–2) and spatial anisotropy (spatial elongation, dataset 3). The patterns were generated by frequency-filtering random noise sequences with a bank of filters of characteristic center spatial and temporal frequencies and spatial orientation bandwidth tiling the spatiotemporal spectrum (Supplementary Fig. 2, Supplementary Movie 2, see Methods).

To characterize tuning for spatiotemporal frequency in orientation selective and non-selective populations, we recorded responses to 30 combinations of center spatial and temporal frequencies (0.02, 0.04, 0.08, 0.16, 0.32 cpd; 0.5, 1, 2, 4, 8, 16 Hz; spatial and temporal bandwidth: 1 octave) (Fig. 1d; dataset 1–2) using spatially isotropic and anisotropic stimuli of two distinct spatial orientation bandwidth (Fig. 1d, left vs. right, infinite vs. 15 deg spatial orientation bandwidth, full width half maximum [FWHM]; dataset 1 and 2). The orientation of the anisotropic stimuli varied at constant speed (45 deg/s; Supplementary Movie 2 center). To probe tuning for spatial elongation, we recorded responses to 16 combinations of spatial frequency and orientation bandwidth (spatial frequency: 0.04, 0.08, 0.16, 0.32 cpd; orientation bandwidth: infinite, 60, 30, 15 deg FWHM; temporal frequency 2 Hz; Supplementary Fig. 2c; dataset 3). The stimuli were presented in a randomized order in 4-s epochs interleaved with 4-s epochs of gray screen. Each combination of spatiotemporal frequency and spatial anisotropy was repeated four times using random noise sequences (see Methods).

A summary of the acquired data is presented in Table 1. We quantified the visual-evoked responses of a total of 87,309 somatic regions located 100–300 μm below the pia (hereafter referred to as 'neurons'), which showed at least one calcium transient during any of the recordings. Unless stated otherwise, the analyses were restricted to neuronal cell bodies showing the combination of high-amplitude visually-evoked calcium responses (stimulus-evoked response amplitude > 3x standard deviation of baseline activity for over one sec) and highly-correlated response time courses across repeated trials (reliably responsive neurons, 75th

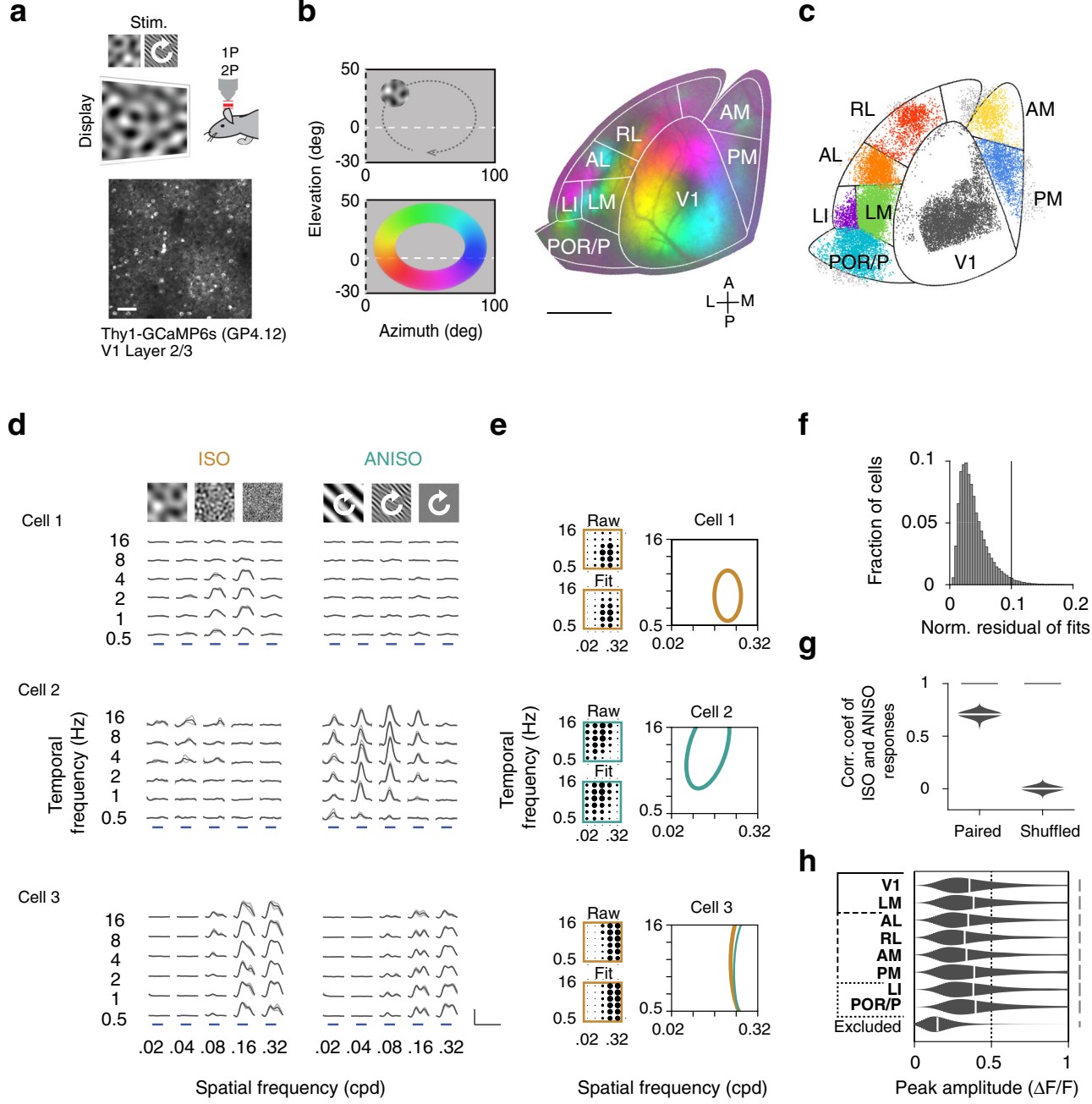

percentile of correlation coefficients between de-randomized stimulus-evoked fluorescence traces across trials, $r > 0.3$, see Methods) (Fig. 1h, Supplementary Fig. 3, 32,580/87,309 cells, dataset 1–3), representing between 25 and 49% of the imaged neurons across visual areas.

**Divergent representations of spatiotemporal frequency in dorsal and ventral visual areas.** In primates, neurons in the dorsal and ventral streams integrate inputs from visual pathways with specific spatial and temporal tuning properties[46,47]. To examine whether a similar architecture exists in the mouse cortex, we quantified the selectivity of responses for spatial and temporal frequency using 2D Gaussian function fits (Fig. 1d–e) (see Methods). From these fits, we extracted response peak amplitudes (Fig. 1h), peak and cutoff spatial and temporal frequencies, spatial and temporal tuning shapes (spatially or temporally lowpass,

bandpass, or highpass) and bandwidth. Good quality fits were obtained for a vast majority of neurons (Fig. 1f, root mean squared error <10% peak amplitude, 26,659/27,701 reliably responsive neurons, Table 1). For each neuron, we examined the data from the fit with higher peak response amplitude (Fig. 1e, thick lines; dataset 1 or 2). The isotropic and anisotropic stimuli yielded highly correlated tuning for spatial and temporal frequency (Fig. 1d, bottom, g, Pearson's correlation coefficients, paired vs. shuffled responses: median values 0.71 vs. 0; 100 random neurons per resampling, 1000 resampling; two-sample KS test, $p < 0.001$, 10,811 neurons responding to both types of stimuli, 10 mice).

Examining the organization of spatiotemporal tuning across visual areas, we observed a high degree of specificity. In individual visual areas, neurons exhibit a broad spectrum of tuning properties (Figs. 2a, c; 3a, c). Across visual areas, specific response patterns emerge (Figs. 2a–d and 3a–d), whereby

**Fig. 1 Characterization of mouse visual cortical areas with spatiotemporal noise stimuli. a** Characterization of visual cortical neurons in primary visual cortex (V1) and higher visual areas (HVAs) in mice. Thy1-GCaMP6s mice (line GP4.12, $N = 10$ mice) were head-fixed during visual stimulation and imaging. Spatially isotropic (ISO) and anisotropic (ANISO) visual noise stimuli of specific spatial and temporal frequencies and orientations were presented to the right visual hemifield (azimuth 0–100 deg; elevation −30–50 deg) (top). While the ISO stimuli had constant frequency spectra, the orientation of the ANISO stimuli was varied slowly to activate neurons with a spectrum of orientation preferences (white arrow). GCaMP6s labeled L2/3 cortical pyramidal neurons in the left visual cortex (bottom) were imaged using 2-photon (2P) microscopy through a cranial glass window. Scale bar: 100 μm. **b** Retinotopic mapping of mouse visual cortical areas. The borders of primary visual cortex (V1) and higher visual areas (HVAs) were identified using 1-photon (1P) widefield calcium imaging of responses to a clockwise circling visual stimulus. Stimulation at different visual field locations (left) activates different regions of the visual cortex (right). This activation pattern (color coded) was used to align the cortical surface to a common area delineation. Scale bar: 1 mm. **c** Distribution of functionally-characterized neurons within the common area delineation. Color-coded dots indicate the estimated neurons' cell body locations and assigned visual areas. Neurons with cell bodies located outside the common delineation (light gray dots) were excluded from the analysis. **d** Example calcium responses to ISO (left) and ANISO stimuli (right) for three simultaneously recorded neurons. The neurons show distinct spatiotemporal selectivity and distinct preferences for ISO and ANISO stimuli (top to bottom). Gray traces and light gray shadows: median ± median absolute deviation (4 trials). Blue bars: 4s stimulus epochs. Scale bars: 1 ΔF/F and 10 s. **e** Model-based estimation of spatiotemporal frequency tuning. Dot plots show average normalized response amplitudes (encoded as dot surface area) as a function of spatial and temporal frequency (upper left) and amplitudes from the corresponding 2-dimensional Gaussian function fits (bottom left). Right panel: Contours show halfmaximum of responses to ISO (yellow curves) and ANISO stimuli (green curves) estimated from model fits. For each neuron, the fit with maximal peak response (thick curves) was selected for further analysis. **f** Histogram shows the distribution of normalized residuals of fits (root mean squared errors divided by the peak amplitude) across the characterized neural population. Good quality fits (<0.1) were obtained for most cells with reliable responses (26,659/27,701 neurons, datasets 1 or 2). **g** Mean Pearson's correlation coefficients of responses to ISO stimuli between paired vs. randomly-shuffled responses to ANISO stimuli (100 cells per bootstrap subsampling, $n = 1000$ iterations). Two-sample KS test, $p < 0.001$. ISO and ANISO stimuli yield similar measurements of spatiotemporal tuning (10,811 neurons responding to both ISO and ANISO stimuli, 10 mice). **h** Peak response amplitudes of selected and excluded neurons. For each distribution in **g** and **h**: kernel density estimator bandwidth 0.05; scale bar: 2.5% cells; black vertical bars: median value.

neurons in distinct visual areas are activated by distinct ranges of spatiotemporal frequencies (Fig. 2b). While V1 neurons respond most strongly to stimuli of low temporal and low spatial frequencies, LM neurons show the strongest activation in response to stimuli of mid-range spatiotemporal frequencies. Similarly, whereas neurons in anterior higher visual areas (AL, RL, AM) show the strongest activation for stimuli of high temporal and low to mid-range spatial frequencies, neurons in posterior higher visual areas (PM, LI, and POR/P) show the nearly opposite response pattern with the strongest activation for stimuli of low temporal and high spatial frequencies. Accordingly, the distributions of tuning shapes (Fig. 2d), peak spatial and temporal frequencies, and spatial and temporal frequency cut-offs show highly significant biases (Fig. 3b, d; Supplementary Fig. 5; hierarchical bootstrap KS tests accounting for inter-animal variations, see Methods). Consistent across animals (Supplementary Fig. 4a–c), the biases are robust to variations in the response reliability threshold used to identify visually-responsive neurons (Supplementary Fig. 4d).

These representational biases also provide indications of visual stream specialization. Ascribed to the dorsal stream, areas AL, RL and AM share a marked preference for fast-varying low-spatial-resolution stimuli. Ascribed to the ventral stream, LI and POR/P share a distinct preference for slow-varying high-spatial-resolution visual stimuli. However, with a preference for slow-varying high-resolution signals, the properties of area PM stand out from those of other dorsal HVAs resembling more those of ventral visual areas. While the differences between AL and PM described here are consistent with the report by Anderman and colleagues (2011)[31], they differ markedly from the data of Marshel et al. (2011)[32], which showed less pronounced differences in spatiotemporal preferences across visual areas (Supplementary Fig. 6, detailed in Discussion).

**Enhanced speed tuning and divergent visual speed representations in higher visual areas.** In primates, the dorsal stream is specialized for visuomotor processing and its neurons are involved in the encoding of visual motion, showing pattern motion selectivity and neuronal responses correlated with visual

motion perception. Although sensitivity for pattern motion is rare in the mouse cortex[33], selectivity for motion speed is observed and may be enhanced in certain HVAs[31,36]. The noise stimuli contain local visual motion, which can be used to examine speed selectivity. For such stimuli, speed is defined as the ratio of the stimulus' center temporal frequency over its center spatial frequency. To study speed representations, we computed from the fits to the responses to isotropic stimuli (dataset 1), a speed tuning index (SI) describing the slant in the neuron's 2-dimensional spatiotemporal tuning[12] (Supplementary Fig. 7). In this quantification, high SI values (SI > 0.5) indicate an approximately invariant speed tuning curve for a range of spatial and temporal frequencies, whereas low ($-0.5 < SI < 0.5$) and negative (SI < −0.5) SI values, in contrast, indicate a lack of such scale invariant speed tuning.

Examining the organization of speed tuning across areas, we found a specific enrichment of speed selectivity in dorsal areas. This enhancement is visible in the gradient of mean SI values across the cortical surface (Fig. 3e). The proportion of speed-tuned neurons is lowest in V1 (25%), which shows low average SI values (Fig. 3e) and an approximately symmetrical spread of SI values around the origin (Fig. 3f). In contrast, anterior dorsal areas comprise higher proportions of speed-tuned cells (AL: 41%, RL: 50%, AM: 47%), with high average speed tuning indices (Fig. 3e) and significant shifts towards large SI values relative to V1 (Fig. 3f, hierarchical bootstrap KS test). The proportion of speed-tuned neurons is also elevated in PM (36%), but the difference is more modest. In comparison, ventral areas LI and POR/P show lower proportions of speed-tuned neurons (LI: 34%, POR/P 33%) and a significant shift towards low SI values relative to dorsal visual areas. The proportion of speed-tuned neurons in LM (38%) appears elevated relative to V1, however, the distributions of SI values for this area mostly did not differ significantly from those observed in other areas.

Examining the organization of preferred (peak) speed, we found again a high degree of specificity (Fig. 3g–h). In comparison to neurons in V1 and LM, neurons in dorsal areas AL, RL, and AM show a distinct preference for higher visual speed (Fig. 3h, right). In ventral areas LI and POR/P, in contrast, neurons show a preference for lower speed. As a result of these

**Table 1 Summary of data.**

| Area | # mice | # cells | # responsive cells (% all cells) | # reliably responsive neurons (% all cells) | | | | | # neurons with good SFTF fits | | |
|---|---|---|---|---|---|---|---|---|---|---|---|
| | | | Dataset 1–3 | Dataset 1 (ISO) | Dataset 2 (ANISO) | Dataset 3 (Elongation) | Dataset 1 \| 2 | Dataset 1 \| 2 \| 3 | Dataset 1 | Dataset 2 | Dataset 1 \| 2 |
| V1 | 8 | 28,617 | 17,103(60) | 6907(24) | 6353(22) | 8352(29) | 9076(32) | 11,313(40) | 6605 | 5926 | 8718 |
| LM | 8 | 15,862 | 10,267(65) | 4887(31) | 5073(32) | 5827(37) | 6865(43) | 7725(49) | 4664 | 4628 | 6593 |
| AL | 7 | 7578 | 4233(56) | 1792(24) | 2188(29) | 1929(25) | 2779(36) | 2961(39) | 1728 | 2037 | 2609 |
| RL | 7 | 8619 | 3722(43) | 1060(12) | 1299(15) | 1362(16) | 1694(20) | 2001(23) | 1042 | 1257 | 1656 |
| AM | 6 | 5052 | 2650(52) | 936(19) | 1076(21) | 1248(25) | 1375(27) | 1654(33) | 922 | 1037 | 1348 |
| PM | 6 | 5802 | 3783(65) | 1388(24) | 1706(29) | 1854(32) | 2168(37) | 2570(44) | 1330 | 1606 | 2096 |
| LI | 6 | 4427 | 2343(53) | 1020(23) | 706(16) | 1009(23) | 1269(29) | 1474(33) | 974 | 664 | 1216 |
| POR/P | 10 | 11,352 | 5595(49) | 1864(16) | 1420(13) | 1585(14) | 2535(22) | 2882(25) | 1770 | 1280 | 2423 |
| Total | | 87,309 | 49,696(57) | 19,854(23) | 19,821(23) | 23,166(27) | 27,701(32) | 32,580(37) | 19,035 | 18,435 | 26,659 |

Table summarizing the number of mice and cells imaged for each area, responsive cells (response amplitude >3sd of baseline ΔF/F₀ for over 1 s for at least one stimulus condition), number and fraction of cells showing highly reliable responses (75th percentile of cross-trial correlation coefficient >0.3) to individual and combinations of datasets and number of cells with good fits (root-mean-squared-error <10% peak amplitude) of the responses of different spatiotemporal frequencies.

biases in tuning, anterior dorsal areas (AL, RL, and AM) increase their activity with increasing stimulus speed (peaking at 100–400 deg/s) whereas ventral areas (LI and POR/P) decrease their activity (peaking at 6–12 deg/s) (Supplementary Fig. 7c, red curves). These biases, which are consistently observed across animals (Supplementary Fig. 7c, red curves), entail a factor of sixteen in variations of mean speed (Fig. 3g, h; hierarchical bootstrap KS test). Interestingly, while measures of preferred speed and speed selectivity are approximately independent in V1, LM, AL, and RL, high SI values in AM, PM, LI, and POR/P seem restricted to lower visual speed (<25 deg/s) (Supplementary Fig. 7d).

Together these data indicate that dorsal and ventral visual areas in the mouse cortex form distinct visual speed representations. Reminiscent of dorsal and ventral areas of the primate visual pathways, the areas show distinct degrees of speed tuning and differentially encode slow and fast stimuli. Compared to AL, RL, and AM, area PM shows a distinct preference for low speed, suggesting it partakes in a separate visual pathway.

**Enhanced encoding of visual speed in higher visual areas.** The speed representations could confer HVAs an advantage in encoding stimuli of different speed. To examine this possibility, we quantified using a neural decoding analysis how accurately the speed of pairs of visual stimuli of similar spatiotemporal frequencies can be discriminated from neuronal activity in mouse visual areas. Specifically, linear support vector machine (SVM) classifiers were used to decode neural activity in response to isotropic spatiotemporal stimuli (dataset 1) and discriminate responses to pairs of stimuli sampled between (Fig. 4a) or along iso-speed lines (Fig. 4d). To evaluate the advantage of HVAs in encoding visual speed, we computed the decoding (discrimination) accuracy for neuron ensembles (pools) of different sizes (Fig. 4b, e, insets) and estimated the fractional differences in decoding accuracy (Δ decoding accuracy) observed for HVA ensembles relative to V1 (Fig. 4b, c) and for stimuli sampled across and within iso-speed lines (Fig. 4e, f).

The analysis revealed enhanced encoding of visual speed in HVAs in comparison to V1 (Fig. 4c, f, Supplementary Fig. 8). Across the dataset, discrimination accuracy increases with neural ensemble size (Fig. 4b, e, insets) showing a dependence on stimulus spatiotemporal frequency (Supplementary Fig. 8a). Differences in the discrimination accuracy between classifiers, however, could be best discerned for small neuron ensembles (Fig. 4b, e). We, therefore, quantified Δ decoding accuracy for a range of spatiotemporal frequencies and a pool size of 16 neurons (Fig. 4c, f). Examining Δ decoding accuracy computed between HVAs and V1, we found that although V1 shows good speed discrimination ability across the frequency spectrum with a maximal accuracy of around 75%, many HVAs show higher decoding performance with peak accuracy at around slopes of average tuning maps and chance-level accuracy centered around the preferred and null frequency (Figs. 2b, 4c, Supplementary Fig. 8b). Examining Δ decoding accuracy computed between speed and iso-speed responses, we found neuron ensembles in dorsal and ventral areas show specific advantages in discriminating stimuli of different speeds relative to stimuli varying along the perpendicular axes (Fig. 4f, Supplementary Fig. 8d). This advantage is most pronounced for areas AL, RL, and AM in the dorsal stream. Nonetheless, ventral areas LI and POR/P also show an advantage for speed coding. Interestingly, areas PM and LM show only a weak advantage over V1. Altogether, the results indicate that HVA neural responses have an advantage for speed coding and this advantage is more pronounced in anterior dorsal visual areas.

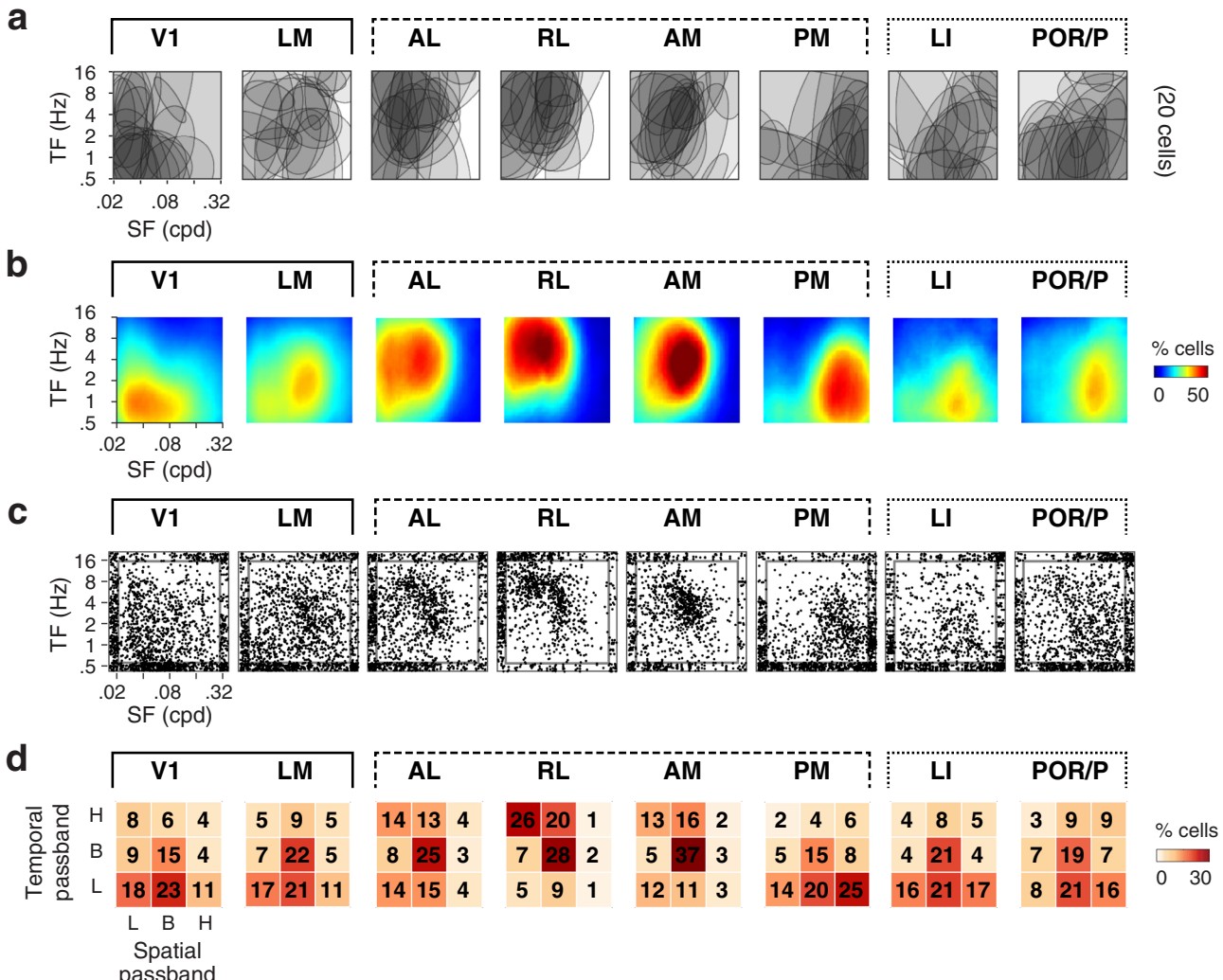

**Fig. 2 Comparison of spatiotemporal frequency representations in V1, LM, and dorsal and ventral visual areas.** For each characterized visual area:
**a** Example spatiotemporal tuning. Overlaid halfmaximum regions of 2-dimensional spatiotemporal tuning for 20 randomly-selected cells. **b** Fraction of activated neurons as a function of center spatial and temporal frequency computed by averaging normalized half-maximum regions of tuning fits. Note how distinct visual areas respond to distinct ranges of spatiotemporal frequencies. **c** Scatter plots showing the distributions of peak spatiotemporal frequencies. Data points scattered at graph edges correspond to data points at the outer limits of spatial and temporal frequencies tested. **d** Fraction of neurons for each spatiotemporal tuning shape (H: high pass. B: band pass. L: low pass).

**Distinct responses to oriented and non-oriented stimuli in ventral and dorsal areas.** We next examined the population representations of oriented and non-oriented spatial features in mouse visual areas. In the primate visual cortex, neurons encode specific orientations[2,48] as well as specific spatial patterns such as the curvature or the elongation of a bar, with more complex spatial patterns overly represented in HVAs[15,49]. In the mouse cortex, neurons show diverse orientation tuning as well as diverse spatial integration properties[22,31,32,50]. These areal differences could reflect processing streams tuned to particular spatial visual patterns. To characterize the organization of spatial tuning across visual areas, we examined the magnitudes of responses to isotropic and anisotropic noise stimuli (datasets 1 and 2) (Figs. 1d and 5a).

Some neurons show a preference for anisotropic, oriented stimuli (Fig. 1d, middle and bottom; 5a, dark and light blue). However, a subset shows the opposite preference, responding preferentially to isotropic, non-oriented stimuli (Fig. 1d, top, 5a, dark gray). To our knowledge, a preference for non-oriented stimuli has never been described and could reflect receptive fields

tuned to non-oriented or weakly oriented features (e.g., curvature). To examine the relative prevalence of these response patterns, we computed an anisotropy preference index (API), defined as the ratio of the difference to the sum of the average responses to anisotropic and isotropic stimuli (see Methods). In this quantification, an API value of 0 indicates responses of similar magnitudes to oriented and non-oriented stimuli whereas API values of −1 or +1 indicate specific responses to isotropic or anisotropic stimuli, respectively. Examining the distribution of API values across the dataset, we found a high prevalence of neurons preferring isotropic stimuli over anisotropic stimuli. About half the cells show comparable responses to both oriented and non-oriented stimuli (49%, 13,168/26,659, −1/3 < API < 1/3). Furthermore, nearly a third of the neurons (31%, 8,211/26,659 cells, datasets 1 and 2) exhibit a preference for non-oriented stimuli or respond exclusively to these stimuli (API < −1/3). In comparison, only a fifth of the cells show a preference for oriented stimuli (20%, 5,280/26,659 cells, API > 1/3).

To determine whether the preference for anisotropic over isotropic stimuli is predictive of the degree of orientation tuning,

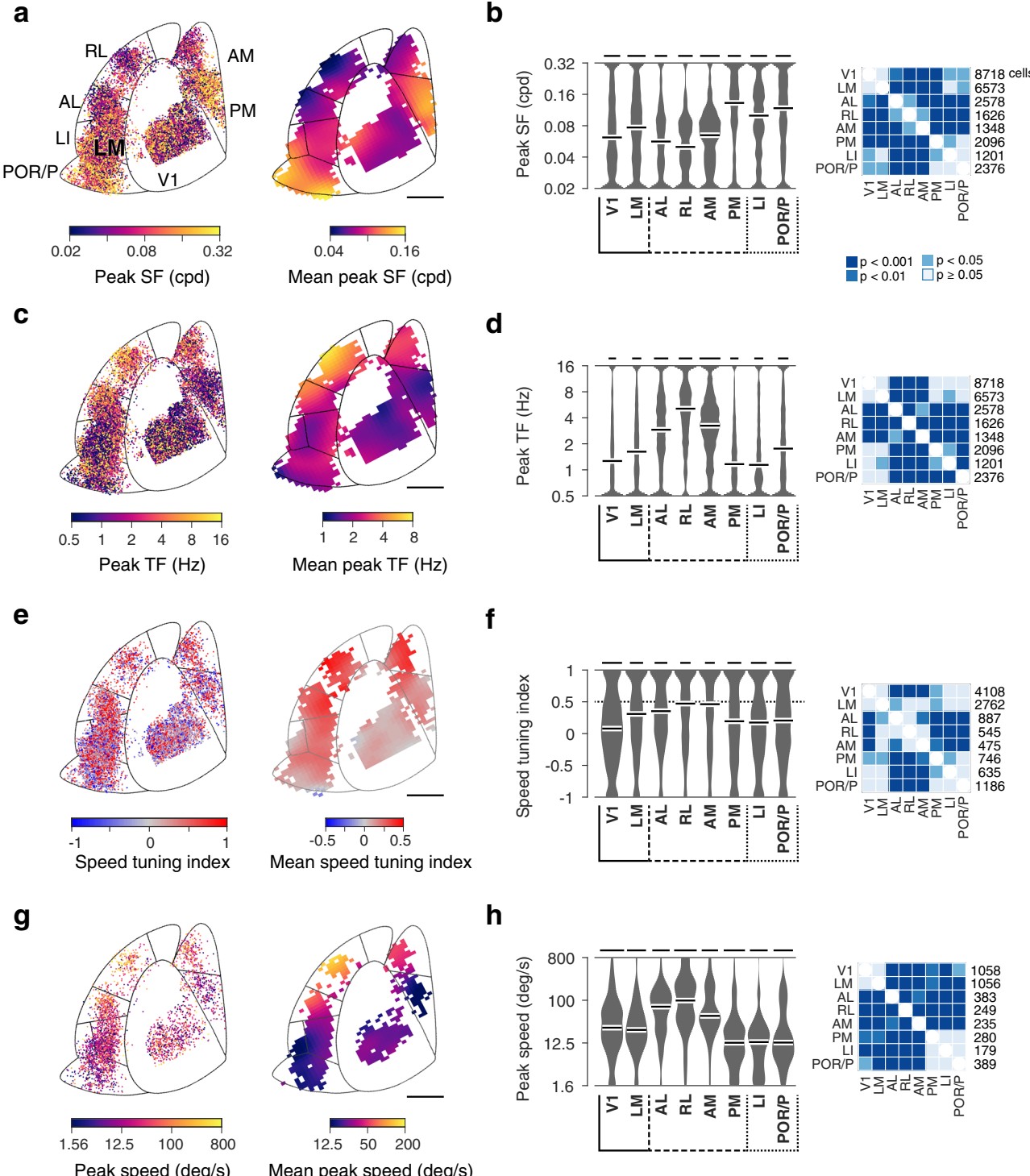

**Fig. 3 Comparison of spatial and temporal frequency preferences, speed selectivity and preference in V1, LM and dorsal and ventral visual areas.**
**a** Distributions of observed spatial frequency preferences across visual cortical areas. Left: color-coded map of peak spatial frequency for individual neurons as function of cortical location. Right: spatially-smoothed map (Gaussian filter width 400 μm). Data across animals (n = 10) presented within the common delineation. Scale bar: 1mm. **b** Left: Probability density functions of observed peak spatial frequency preferences for each visual area. Kernel density estimator bandwidth 0.05; scale bar: 2% cells; white bars: median value. Right: two-sided hierarchical bootstrap KS tests assessing statistical significance of differences in density functions across visual areas accounting for inter-animal variability; see details in Methods. **c**, **d** Distributions of peak temporal frequency preferences across areas, same format as **a** and **b**. **e** and **f** Distributions of speed tuning index (SI) across areas, same format as **a** and **b**. The dashed line indicates the selection threshold for speed-tuned cells (SI > 0.5, dataset 1). **g**, **h** Distributions of peak speed of speed-tuned cells, same format as **a** and **b**. SF: spatial frequency. TF: temporal frequency. Scale bars in **d**, **f**, **h** represent 1.5, 1, 0.5% cells, respectively.

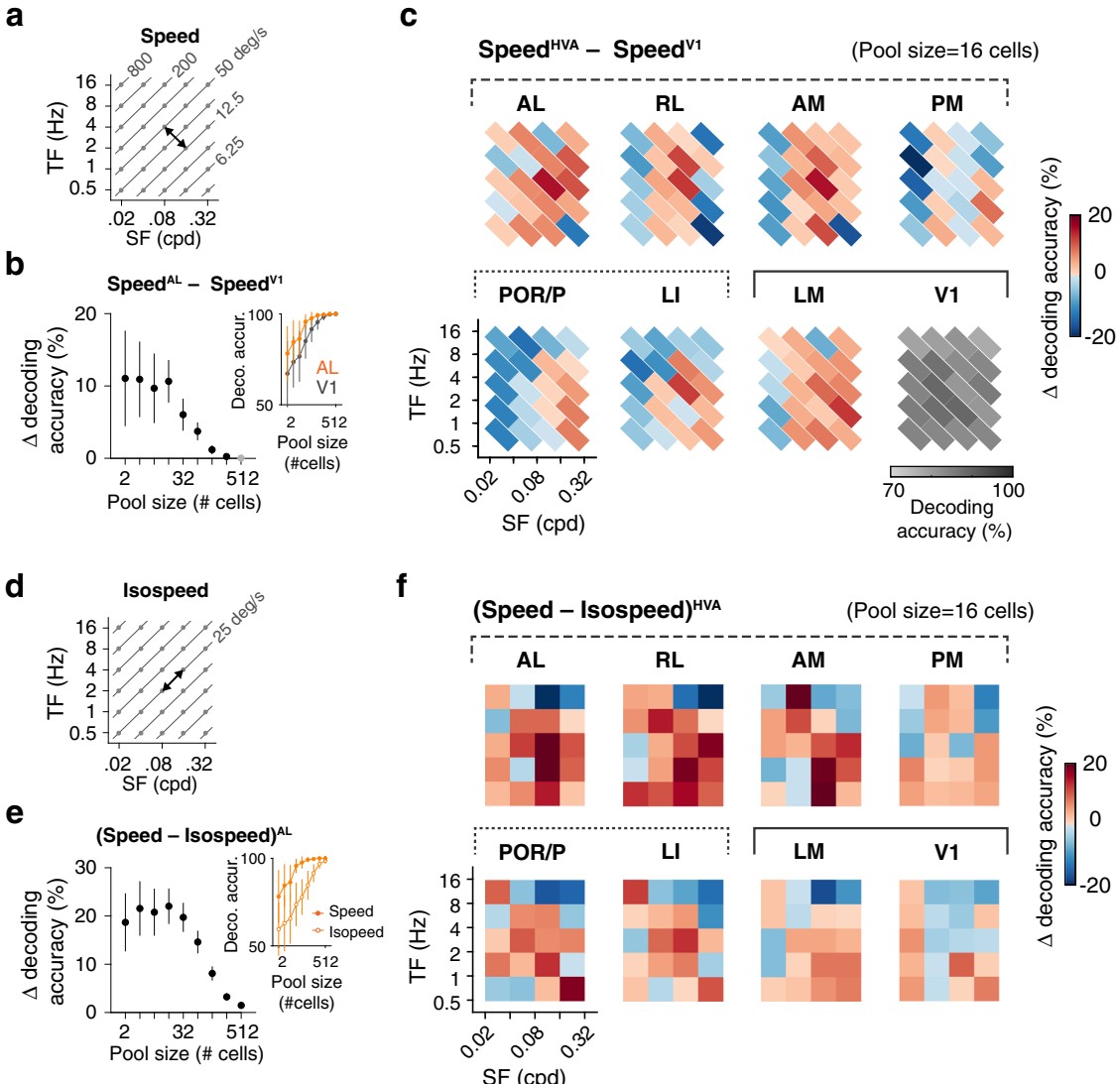

**Fig. 4 Neural decoding reveals enhanced speed discrimination in higher visual areas. a** Decoding neural responses to spatiotemporal stimuli of different speed (speed pairs). Speed pairs are sampled orthogonally to iso-speed lines and have a 4-fold difference in speed. **b** Fractional difference of decoding accuracy in area AL relative to V1, for the speed pair shown in **a**. The difference varies as a function of the size of the decoding neuron ensemble (pool size). The curve is calculated based on the speed decoding performance of AL and V1 data (inset, the chance level at 50% accuracy; $n = 50$ randomizations; mean ± 95% confidence intervals). Two-sample t-tests. Significant differences are shown as black markers, otherwise gray markers. **c** Summary of differences in speed decoding performance in higher visual areas relative to V1. V1 speed decoding performance is shown as a reference. See Supplementary Fig. 8b for statistical significances. Pool size: 16 cells. **d** Decoding neural responses to stimuli of different spatiotemporal frequencies but identical visual motion speed (iso-speed pairs). **e** Fractional difference of decoding accuracy observed when comparing results for the speed and iso-speed pairs shown in **a** and **d**, in area AL. Note the increased decoding performance for the speed pair relative to the iso-speed pair. Solid circle: speed pair. Open circle: iso-speed pair. Same style as **b**. **f** Summary of differences in decoding accuracy between speed pairs and corresponding iso-speed pairs in different areas. See Supplementary Fig. 8d for statistical significances. Pool size: 16 cells.

we computed, from the time courses of responses to anisotropic stimuli (Fig. 5a, dark blue and gray, dataset 2), approximate orientation tuning curves and associated orientation selectivity indices (OSI) quantifying the degree of orientation tuning (Fig. 5c–e; Supplementary Figs. 10, 11; see Methods). We observed a correlation between the neurons' preferences for isotropic and anisotropic stimuli and their degree of orientation selectivity. While neurons with high API values show high OSI values (Fig. 5f), indicative of a highly orientation-tuned population, neurons with low API values show comparatively low OSI values (Fig. 5f), indicating weaker selectivity.

Previous studies reported that the degree of orientation tuning of cortical neurons varies across mouse visual areas[31,32]. To determine whether the proportions of neurons tuned to oriented and non-oriented stimuli covary across visual areas, we examined the distributions of API and OSI values as well as the proportion of neurons responding exclusively to isotropic stimuli. We observed a non-uniform distribution of neurons tuned to oriented and non-oriented stimuli in areas of the dorsal and ventral visual streams (Fig. 5b, g). While ventral visual areas LI and POR/P exhibit marked shifts towards low API values relative to other visual areas (Fig. 5b, hierarchical bootstrap KS tests) and

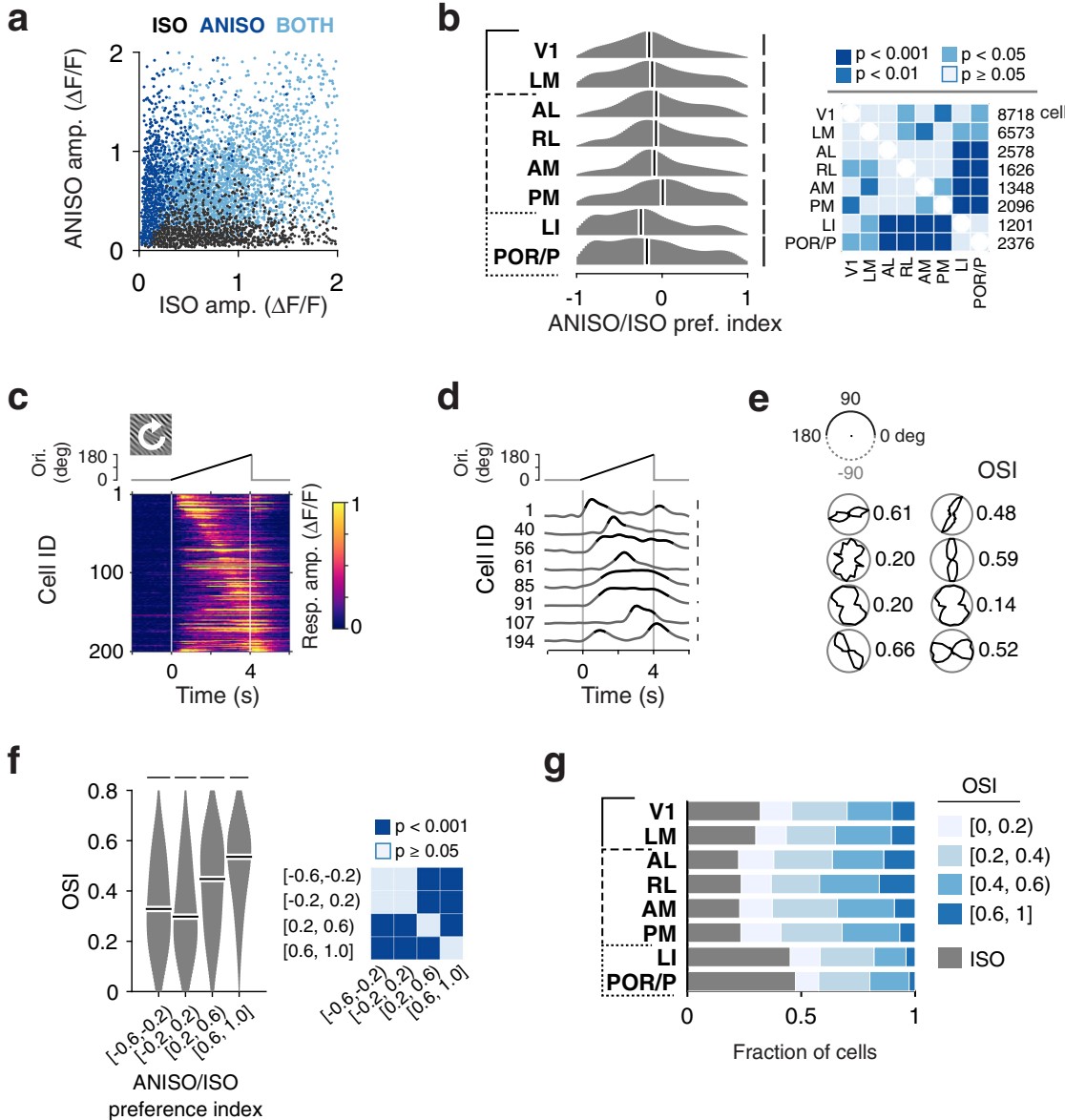

**Fig. 5 Diverse responses to oriented and non-oriented stimuli in ventral and dorsal visual areas. a** Scatter plot showing diverse amplitudes of visual cortical neurons' responses to isotropic (ISO) and anisotropic (ANISO) stimuli (datasets 1–2). Dark blue, gray, and light blue dots represent cells showing reliable responses to ANISO stimuli only, ISO stimuli only, and to both types of stimuli, respectively. **b** Quantification of neurons' ANISO/ISO preference for populations in each area. The ANISO/ISO preference index (API) is defined as the ratio of the difference in peak responses to the stimuli over their sum. For each distribution: Kernel density estimator bandwidth 0.1; scale bar: 2% cells; white bars: median value. Right panel: two-sided hierarchical bootstrap KS tests for pairwise comparison. **c** Calcium traces of 200 randomly-selected V1 neurons in response to ANISO stimuli of time-varying orientation (icon, top inset). In these recordings, time to response peak and response dynamics can be used to estimate orientation preference and selectivity, respectively. Vertical lines at 0 and 4 s indicate onset and offset of visual stimulus. The top inset indicates the time-varying orientation of the visual stimulus. **d** Subset of the data shown in **c** plotted as traces. The part over 50% peak amplitudes in each trace is shown in black. Scale bar: 0.3 ΔF/F. **e** Approximate orientation tuning curves estimated from the response time courses shown as polar plots, for example neurons in **d** (left to right, top to bottom). Black curves show peak-normalized calcium response amplitudes to different orientations. The upper half of the polar plot is duplicated, rotated, and shown in the lower half, presented together to add visual ease. Estimated orientation selectivity index (OSI), quantified as (1–circular variance), is indicated for each neuron (details in Methods, Supplementary Fig. 11). **f** Distributions of OSI values for cells with distinct ranges of ANISO preference index. Neurons with high API values show high OSI values indicative of sharp orientation tuning, whereas neurons with low API values show lower OSI values indicative of weaker orientation selectivity. Scale bars: 2.5% cells. Two-sided hierarchical bootstrap KS tests (right panel). **g** Fraction of neurons showing different orientation selectivity in different areas.

elevated proportions of cells responding exclusively to non-oriented stimuli (Fig. 5g, gray), dorsal visual areas AL, RL, AM, and PM show low proportions of neurons responding exclusively to non-oriented stimuli (Fig. 5g, gray) and high proportions of orientation-tuned neurons (Fig. 5g, dark blue). In comparison, V1

and LM lie midway with API and OSI values between ventral and dorsal visual areas. Together, these results indicate that oriented and non-oriented stimuli are represented broadly across visual areas and distributed specifically across dorsal and ventral visual streams.

**Discrete response types underlie divergent representations of oriented and non-oriented stimuli**. To investigate the spatial tuning properties underlying the preferences for oriented and non-oriented stimuli, we characterized, in a separate dataset (dataset 3; 22,816 cells showing reliable responses, 10 mice; Table 1), the responses to visual patterns of varying spatial scale and elongation (Fig. 6a) using noise stimuli comprising four grades of orientation bandwidth (orientation bandwidth: infinite, 60, 30, 15 deg FWHM) and four spatial frequencies (spatial frequency: 0.04, 0.08, 0.16, 0.32 cpd; temporal frequency: 2 Hz) (Fig. 6b). Through their rotation, the anisotropic stimuli covered a full range of orientation (Fig. 6a, white arrows) allowing activation of neurons with diverse orientation preferences. The joint tuning of responses for spatial frequency and orientation bandwidth yielded separable response patterns (Fig. 6b, top), which could be quantified by examining the amplitudes of responses as a function of elongation at the neurons' peak spatial frequency (Fig. 6b, bottom). We grouped neurons according to their ANISO/ISO preferences (API values) and the similarity of their tuning for elongation and examined the response time courses and OSI for distinct ranges of API values.

The experiments revealed three broad response types (Fig. 6c, d). Neurons with low API values ([−1,−1/3]) show strong responses to isotropic stimuli and a steep decrease in response amplitude with spatial elongation (Fig. 6b, cell 1; 6c, top). These cells had too weak responses to elongated stimuli to probe orientation tuning (Fig. 6d, top), so we termed them non-orientation selective neurons (non-OS cells). In contrast, neurons with high API values ([1/3, 1]) show weak or no responses to isotropic stimuli and strong responses to elongated stimuli (Fig. 6b, cell 2; 6c, middle). In comparison to non-OS cells, these neurons show sharp calcium transients and sharp orientation tuning (Fig. 6d, middle, Supplementary Fig. 12a), so we termed them sharply orientation selective neurons (sharp-OS cells). Finally, neurons with mid-range API values ([−1/3,1/3]) show robust, comparable responses to both isotropic and anisotropic stimuli (Fig. 6b, cell 3; 6c bottom). In comparison to sharp-OS neurons, these cells have significantly broader orientation tuning, hence termed broad-OS cells (Fig. 6b, cell 3; 6c bottom; Supplementary Fig. 12a; dataset 3; sharp-OS vs. broad-OS: median OSI 0.3 vs. 0.5; 12,143 vs. 7313 neurons; two-sample KS test, $p < 0.001$).

Applying ward linkage clustering to the data, we observed non-uniform representations of sharp-OS, broad-OS cells, and non-OS cells across lower and higher visual areas (Fig. 6e, f). While ventral areas LI and POR/P show a high proportion of non-OS neurons and a low proportion of sharp-OS neurons, dorsal areas AL, RL, AM, and PM show the opposite pattern, a high proportion of sharp-OS neurons and a relative scarcity of non-OS neurons. The distributions of tuning types in V1 and LM are midway between ventral and dorsal areas. However, LM comprises a higher proportion of sharp-OS cells relative to V1. These differences in the relative frequency of response types in visual areas are consistent with differences in average tuning for elongation observed between areas, which were robustly observed across mice (Supplementary Fig. 12b). Together these data indicate that oriented and non-oriented stimuli are encoded in the activity of discrete cell groups which are differentially represented across cortical visual areas and across visual streams.

**Clustering analysis reveals the organization of parallel processing streams**. Finally, we examined the possibility of functional cell groups tuned to both spatial and temporal features. We applied non-supervised clustering (k-means) to the neurons' responses to isotropic and anisotropic stimuli (datasets 1 and 2) (Fig. 7, Supplementary Fig. 13; Methods). The analysis yielded 12 minimally-distant clusters which are broadly represented across visual areas (Fig. 7a; Supplementary Fig. 13a). These clusters capture the range of the neurons' tuning properties to the stimuli (Fig. 7b–d), showing characteristic preferences for spatiotemporal frequency, isotropic and anisotropic stimuli (Fig. 7b). They also differ in tuning for speed, orientation, and spatial elongation (Fig. 7c, d). A 2-dimensional representation of the dataset (t-SNE) shows the clusters uniformly tile the response space with no clear separation (Fig. 8b). Thus, neurons of diverse spatiotemporal properties lie along a continuum rather than forming discrete types.

Despite the lack of discrete cell groups, it is clear that each area features a broad diversity of responses, each with its own functional cellular makeup. Hierarchical bootstrapping of response type distributions shows robust, significant differences across visual areas (Fig. 8c). The pairwise distances between distributions of response types show both a similarity and divergence of representations within and between dorsal and ventral visual streams (Fig. 8d). Specifically, ventral and anterior dorsal areas show nearly opposite distributions of tuning preferences and, amongst themselves, show highly similar functional makeups (Fig. 8d; Supplementary Fig. 14). Ventral areas LI and POR/P show the most similar functional cellular makeup. Anterior dorsal areas AL, RL, and AM show more similar albeit still distinct distributions of properties. However, PM shows tuning preferences that are opposite to the properties of other dorsal areas. Furthermore, it shows the highest cell density in regions of the spatiotemporal tuning that are poorly represented in LI and POR/P (Fig. 8a) and is significantly different from the ventral areas in terms of the cellular makeup (Supplementary Fig. 14, hierarchical bootstrapping, PM-PM vs. PM-ventral areas, Cohen's $d > 2$), indicating it may form its own distinct visual information substream.

## Discussion

By investigating with 2-photon calcium imaging in the mouse the visual response properties of thousands of cortical excitatory neurons, we characterized the functional organization of eight visual cortical areas and investigated the functional specialization of associated cortical processing streams. Using noise stimuli, we found neurons in the superficial cortex carry highly-diverse and highly-specific visual representations, differing in spatiotemporal frequency tuning, stimulus speed selectivity, and sensitivity to oriented and non-oriented stimuli. While V1 and LM neurons show broadly dispersed tuning properties, HVA neurons are more specific, representing narrower ranges of spatiotemporal frequencies (Figs. 2 and 3) and showing more specific biases for stimulus speed and spatial anisotropy (Figs. 4–6). Ventral and dorsal HVAs show distinct tuning properties, responding differentially to oriented and non-oriented stimuli (Figs. 5 and 6) and encoding complementary ranges of spatial and temporal frequencies, and speed (Figs. 2–4). While responses to stimuli of varying orientation bandwidth and elongation fall into discrete clusters (sharp-OS cells, broad-OS cells, and non-OS cells) (Fig. 6), the joint spatial and spatiotemporal tuning lies on a continuum (Figs. 7 and 8). Finally, a direct comparison of visual representations across visual areas (Fig. 8) indicates that the mouse visual cortex can be parcellated into at least three distinct functional streams, which run through ventral areas, anterior dorsal areas, and medial area PM, respectively. These functional results provide a detailed map of spatial and spatiotemporal tuning properties, which complements the anatomical evidence of parallel visual processing streams in the mouse cortex[21,51,52].

Previous studies had reported distinct spatiotemporal tuning properties in AL and PM[28,29,31]. This study extends these

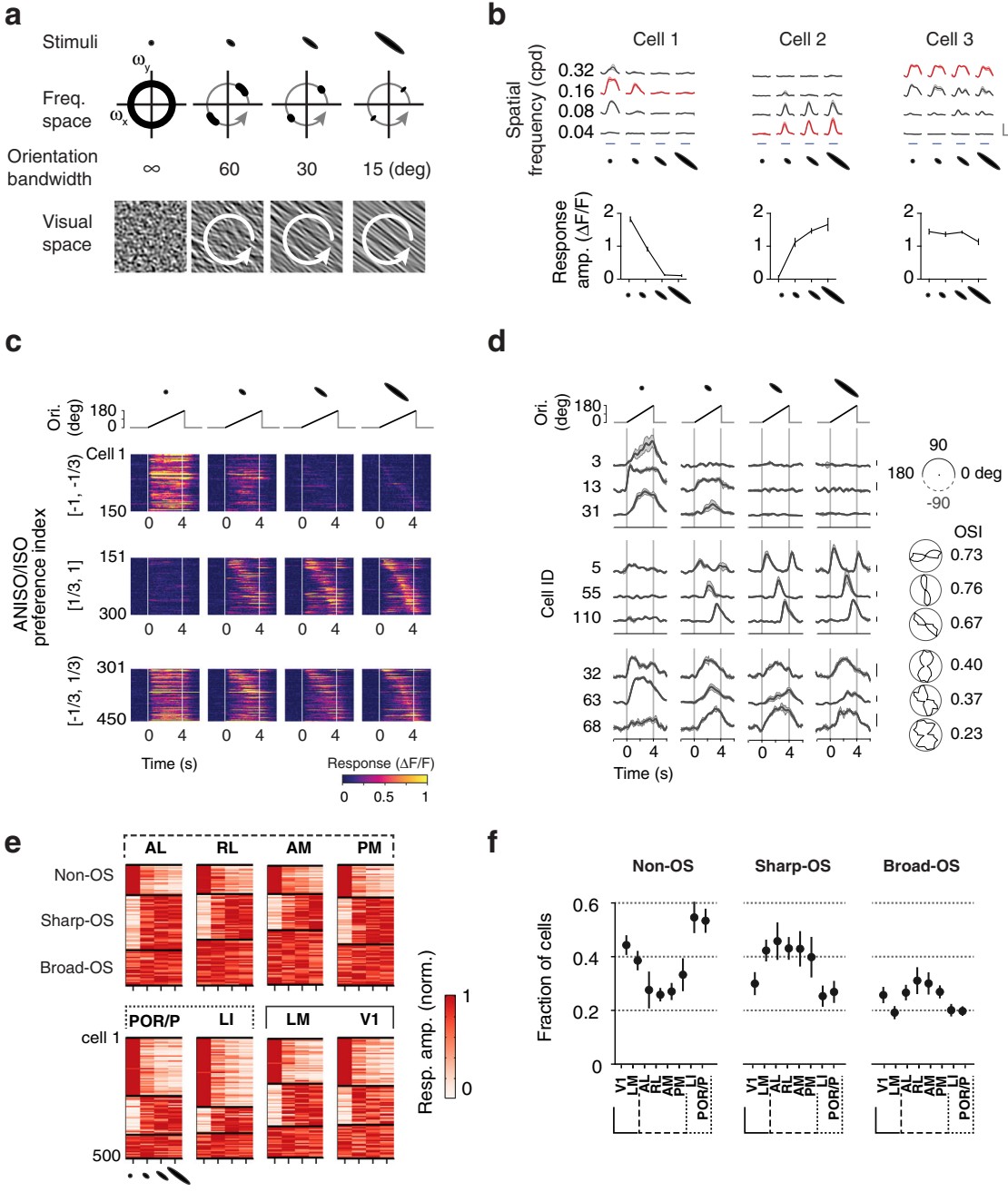

**Fig. 6 Discrete response types underlie divergent representations of oriented and non-oriented stimuli. a** Stimuli to investigate the tuning for spatial elongation in visual cortical populations. Varying the orientation bandwidth (top) changes the correlation of orientation over space and hence the elongation of visual patterns (bottom). We recorded responses to a full matrix of stimuli comprising four center spatial frequencies (orientation bandwidth: infinite, 60, 30, 15 deg FWHM) and four orientation bandwidth (center spatial frequency: 0.04, 0.08, 0.16, 0.32 cpd; center temporal frequency 2 Hz) (dataset 3). The orientation of the anisotropic stimuli varied over time within each stimulus epoch (arrows, 45 deg/s). **b** Example responses illustrating the types of responses observed: isotropic and anisotropic preferring responses (cell 1 and 2) and elongation invariant responses (cell 3). Same cells as in Fig. 1d. The tuning curves (bottom, n = 4 trials) were measured at the neuron's peak spatial frequency (top, red traces). Scale bar: 1 ΔF/F and 10 s. Mean ± s.e.m. **c** Heat maps showing example time courses of population responses to stimuli of four distinct orientation bandwidth. Each block shows responses of 150 randomly-selected cells simultaneously-imaged in V1, grouped by ANISO/ISO preference indices (dataset 3, ranges [−1,−1/3], [1/3,1], and [-1/3,1/ 3]), calculated from the responses to the stimuli with infinite and 15-deg orientation bandwidth. Within blocks, rows (time courses) are sorted according to time to response peak, measured from the 15-deg orientation bandwidth condition (rightmost column). **d** Calcium traces (left) and polar plots (right) of a random subset of neurons in **c**. Scale bar: 0.3 ΔF/F. **e** Color-coded maps of random samples of elongation tuning curves showing the distinct proportions of tuning types in visual areas (500 neurons per area). The three clusters obtained with ward linkage clustering correspond to non-OS, sharp-OS, and broad-OS (OS: orientation selective). In these plots, each row corresponds to one neuron, and each column corresponds to one stimulus condition. Note the uneven representations of the clusters across visual areas. **f** Fraction of cells belonging to each cluster in each visual area. Means and s.e.m. of the fractions were estimated using hierarchical bootstrapping (n = 10 subsampling, 500 random neurons per area per subsampling).

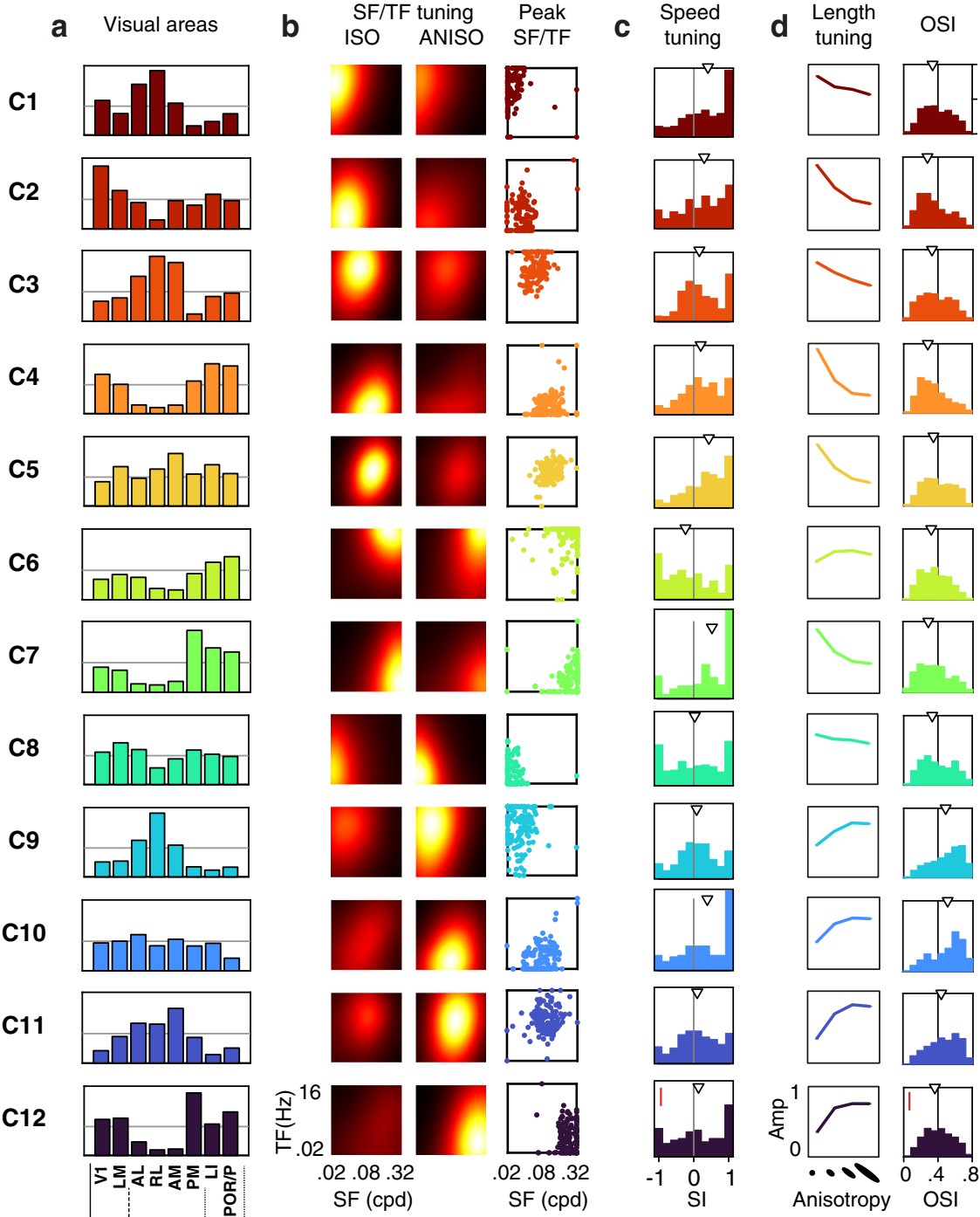

**Fig. 7 Unsupervised clustering analysis uncovers the diversity of tuning underlying specialized representations in mouse visual areas. a** Non-uniform representations of functional clusters in response to isotropic (ISO) and anisotropic (ANISO) stimuli. Each plots shows the proportions of neurons belonging to one particular functional cluster for each visual area. Unsupervised k-means clustering of responses to noise stimuli yielded 12 functional clusters (dataset 1 & 2, see Methods and Supplementary Fig. 13 for details). The horizontal lines correspond to 10% of cells in each visual area. Proportions obtained from random sampling of 2000 neurons per area. **b** Distinct spatiotemporal tuning properties across clusters. Left: heat maps show the average of peak-normalized fits of the responses to ISO and ANISO stimuli. Right: scatter plots show the distribution of peak frequencies. **c** Distinct speed tuning properties across clusters. **d** Distinct orientation tuning properties across clusters. Left: average tuning curves for stimuli of distinct orientation bandwidth (elongation), showing consistency with the preferences for ISO or ANISO stimuli in **b**. Right: distribution of OSI obtained from dataset 3. Arrowhead: median values. SF: spatial frequency. TF: temporal frequency. SI: speed selectivity index. Red scale bars in **c** and **d**: 10% cells.

observations to a broader set of visual areas using a broader set of visual stimuli. Previous large-scale studies of HVAs measured 1-dimensional tuning at a fixed spatial or temporal frequency[32,37], which can be suboptimal for measuring neurons' spatial and temporal selectivity and cannot capture space-time inseparable response patterns such as selectivity for stimulus speed. Focusing on cortical excitatory neurons, using stimuli covering the full spatio-temporal frequency spectrum, we found the preferred spatial and temporal frequencies can differ by as much as 8 fold between visual areas (Fig. 3b, d, h), differences much greater than previously

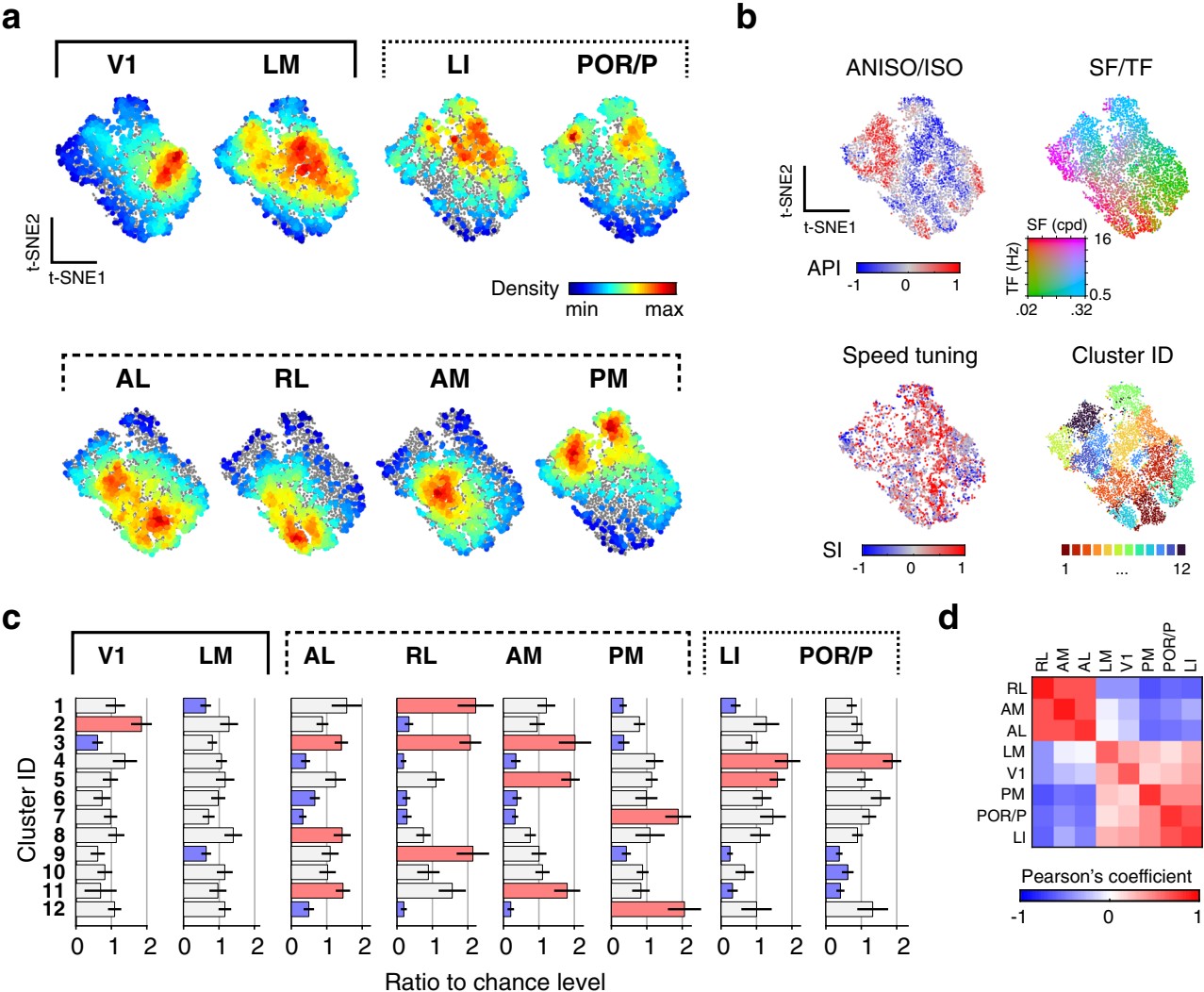

**Fig. 8 Diversity of tuning underlying parallel processing streams. a** t-SNE representations of the diversity of joint spatial and spatiotemporal tuning in response to isotropic and anisotropic stimuli within and across visual cortical areas (dataset in Fig. 7). Colored dots show the density of cells of particular tuning for each visual area (2000 randomly sampled neurons per area). Full dataset shown as gray dots in the background for reference. **b** Functional properties of neurons in the t-SNE representation, showing each cell's ANISO/ISO preferences index (API, top left), peak spatial and temporal frequencies (top right), speed tuning indices of neurons (SI, bottom left), and cluster affiliation (bottom right). **c** Bar plots show the representations of the clusters within each area, quantified as the ratio of clusters relative to chance level (1/12). Over- and underrepresented clusters are shown in red and blue, respectively; otherwise, gray. Mean ± s.e.m. (hierarchical bootstrapping, $n = 1000$ random repetitions). **d** Pearson's correlation coefficient of cluster representations between visual areas. Note the anticorrelated representations of tuning in anterior dorsal and ventral areas (AL, AM, RL vs. LI and POR/P). Note also how the tuning representation in area PM differs from that of other dorsal areas.

appreciated[32] (Supplementary Fig. 6). The discrepancy could reflect differences in experimental conditions such as the calcium indicator employed and the use of anesthetics during the recordings[32]. While the spatiotemporal tuning preferences we observed in AL and PM are generally consistent with results from studies in awake animals[29,31,37] (Supplementary Fig. 6), we observed weaker speed tuning in PM than in AL which seems to contradict previous studies[31,36]. The differences could reflect our use of incoherent motion noise stimuli instead of drifting gratings to measure speed selectivity. The stimuli differ in degree of motion coherence, which could influence estimates of speed selectivity[39].

While the results suggest selectivity for spatial and temporal frequency is an important driver of HVA specialization, an open question is whether the spatiotemporal biases observed are a property of the visual areas or a consequence of their biased retinotopic representations[19]. Although we cannot rule out a contribution of biased retinotopic representations, we note that

only highly visually responsive neurons with receptive fields on the visual display were included in the analysis. We also note the close correspondence between area-specific tuning preferences and the retinotopic borders that were measured independently (Fig. 3a, c, e, g), as well as the relative uniformity of spatiotemporal preference within visual areas. Together, this suggests the observed biases in spatiotemporal preferences are not entirely driven by the biased retinotopic representations.

The properties of sharp-OS cells and broad-OS cells we identified are in line with previous studies; however, the properties of non-OS cells are surprising (Figs. 5 and 6). While mouse V1 L2/3 excitatory neurons were reported to be exclusively orientation tuned[22,38], we found that between 40 and 60% of excitatory neurons in V1, LM, and ventral areas respond much stronger to spatially isotropic stimuli than to anisotropic stimuli. The suppression of activity in response to elongated stimuli is an indication of nonlinear spatial integration. A linearly integrating

neuron with an untuned receptive field should show responses of equal or similar strength to oriented and non-oriented stimuli of the same spatiotemporal scale. One possible mechanism could be that non-OS cells are strongly shaped by inhibitory surround mechanisms (surround suppression). Responding robustly to spatially-extended isotropic stimuli but not to elongated aniso-tropic stimuli, non-OS cells may, in particular, be affected by orientation-tuned suppression, which has also been observed in the mouse visual cortex[50,53]. Because of their specific spatial integration properties, non-OS cells could support specific visual processing such as boundary identification, shape coding, and scene analysis[15,49,54]. Computational modeling of efficient encoding of natural image structures predicts highly diverse feature detectors, including classic Gabor-like edge detectors and equally abundant detectors for non-oriented spatial patterns[55,56]. These model receptive fields resemble neurons' receptive fields probed with natural images in the visual cortex of primates and cats[57,58] and could potentially explain non-OS cells' preferences for non-oriented noise (Fig. 6).

Our results provide functional evidence of multiple parallel processing streams in the mouse visual cortex. Ventral areas LI and POR/P preferentially encode non-oriented features and slowly-varying high-resolution visual inputs. In contrast, anterior dorsal areas AL, RL, and AM show pronounced responses to visual motion, preferentially encoding oriented stimuli and fast varying visual inputs of low-to-mid spatial frequencies. These functional properties could reflect ventral and dorsal processing streams with distinct roles in shape and motion processing.

Interestingly, our data suggest that the area PM, which is tra-ditionally ascribed to the dorsal stream based on its location and connectivity[21,51,52], might form a separate processing stream or substream. Unlike ventral areas, PM shows high orientation selectivity; unlike anterior dorsal areas, PM shows nearly opposite preferences for spatiotemporal frequency and functional cellular makeups, and weak speed selectivity. Supporting this notion, recent fine-scale connectivity studies revealed segregated input pathways to PM and anterior dorsal areas[27–30], and relatively weak connection between them. PM also shows specific responses to coherent visual motion[39], and is a primary source of visual inputs to the retrosplenial cortex[21], which links to navigation and contextual learning[59,60]. Hence, PM might reflect a specialized processing stream in the mouse. It remains to be tested if further branching of processing streams also exists in other species.

While the primate ventral and dorsal streams are broadly segregated for object recognition and action guidance[8], basic visual features, such as the spatial and temporal scales of visual stimuli, orientation, and motion, are represented in both streams and necessary for different aspects of perception and behaviors. Dorsal areas estimate an object's form and motion in conjunction to enable successful interactions[61]. Ventral areas identify objects using information about the spatial features, as well as the relative motion to the background[62]. Similarly, in the mouse, dorsal areas might use the integrative encoding of visual motion and location signals to guide navigation and interaction with the environment[24,31,33,39,63]. Mouse ventral areas bear specialized visual selectivity and connectivity that imply a role in identifying and pursuing small objects, such as moving crickets or overhead threats[40,41,51,64–66]. Other higher-order ventral-like functions (e.g., object recognition) in the mouse or in the rat, albeit reportedly existent[67,68], might not be as developed as those in higher visual mammals. In sum, these similarities and differences between the mouse and the primate may reflect evolutionarily conserved and species-specific adaptation of visual cortical organization and functions.

The specificity and diversity of functional cell types observed within and across cortical areas are in line with the known

anatomy of intercortical and thalamocortical connectivity. On the one hand, functionally similar areas tend to receive inputs from common sources and show strong recurrent connectivity, whereas functionally divergent areas receive segregated inputs from V1 and the thalamus, and are less interconnected[21,27–29,41,52]. On the other hand, although rela-tively weak, the anatomical connectivity between dorsal and ventral, anterior and posterior areas is also extensive[21,52], how-ever, the functional properties of the underlying intracortical projection neurons remain uncharacterized. The shared tuning types observed between functionally divergent areas might form specialized long-range connections, and this possibility remains to be tested. Future studies combining in vivo cellular physiology along with circuit mapping and manipulation should provide important insights into the organization rules of connectivity and information flow between visual cortical areas.

In conclusion, the current study provides a comprehensive characterization of tuning properties in mouse visual cortical areas, addressing joint spatial and spatiotemporal properties and providing insights on the encoding of spatial and temporal information, functional cellular makeups, distributed specialized representations, and organization of parallel processing streams. The results underscore the richness of visual cortical repre-sentations and can serve as an anchor point for future studies on visual processing and behavior, cortical and thalamic con-nectivity, and biologically relevant computational models.

## Methods

**Animals and surgery.** All experiments were approved by the Animal Ethics Committee of KU Leuven. C57BL/6J-Tg (Thy1-GCaMP6s)GP4.12Dkim/J mice[43] between 2 and 3-month of age (5 male and 5 female) were used for chronic widefield and cellular calcium imaging experiments[69]. The mice were single-housed in an enriched environment (cotton bedding, wooden blocks, and running wheel), with 12 h–12 h light-dark cycle, 19–21 degrees cage temperature, 30–70% humidity. Standard craniotomy surgeries were performed[42]. The mice were anes-thetized with isoflurane (2.5–3% induction, 1–1.25% surgery). A custom-made titanium frame was attached to the skull, and a glass cranial window was implanted in the left hemisphere over the visual cortex. In five mice, a 5-mm cranial window was centered over V1 (3.10 mm lateral from lambda, 1.64 mm anterior of the lambdoid suture). In the other five mice, a 4 mm wide glass window was placed over the posterior temporal cortex (3.80 mm lateral from lambda, 1.64 mm anterior of the lambdoid suture). Buprenex and Cefazolin were administered post-operatively (2 mg/kg and 5 mg/kg respectively; every 12 h for two days). All mice were habituated to human handling and the imaging setup at least three days before data acquisition. During the imaging experiments, the mice were comfor-tably head restrained on a treadmill.

**Visual stimulation.** A 22-inch LCD monitor (Samsung 2233RZ, 1680 by 1050-pixel resolution, 60 Hz refresh rate, brightness 100%, contrast 70%) with a mean luminance of 54–80 cd/m$^2$ (from edges to center) was positioned 18 cm in front of the right eye, covering 100 by 80 degrees of the visual field (0 to 100 deg in azimuth from central to peripheral and −30 to 50 deg in elevation from lower to upper visual field). Visual stimuli were generated in MATLAB (The Mathworks, Natick, MA) and presented using PsychoPy2 and custom Python code. A spherical geo-metric correction was applied to the stimuli to define eccentricity in spherical coordinates. The stimuli were generated by applying narrow spatiotemporal bandpass filters (spatial and temporal bandwidth: 1 octave) to random noise (1/f noise) (Supplementary Fig. 2; Supplementary Movie 2) and varying center spatial and temporal frequencies and orientation bandwidth. For the retinotopic mapping, a circular patch of bandpass filtered noise (20 deg in diameter, 0.08 cpd, 2 Hz) moved clockwise along an elliptic trajectory on the display (azimuth: 10 to 90 deg; elevation: −20 to 40 deg; 20 s per cycle with 20 repetitions; Fig. 1b, Supplementary Fig. 1; Supplementary Movie 1). For dataset 1 and 2, the filters included 30 combinations of center spatial and temporal frequencies (spatial frequency: 0.02, 0.04, 0.08, 0.16, 0.32 cpd; temporal frequency: 0.5, 1, 2, 4, 8,16 Hz) for 2 orientation bandwidth (infinite and 15 deg full-width-at-the-maximum or FWHM). For dataset 3, the filter set included 16 combinations of center spatial frequencies and orientation bandwidth (0.04, 0.08, 0.16, 0.32 cpd; infinite, 60, 30, 15 deg FWHM) generated at a fixed center temporal frequency (2 Hz). The center spatial orien-tation of the stimuli with finite bandwidth was varied to rotate clockwise at a rate of 45 deg/second. The stimuli had 50% standard root-mean-square contrast. Each stimulus consisted of 4 s of filtered noise which was presented in alternation a 4-s epoch gray screen of the same average luminance. The stimuli were presented in a randomized order in four pseudorandomized trials. For each trial, a different set of

seeds were used to generate unique noise stimuli, with varying phases yet constant frequency spectra across trials.

**1-photon and 2-photon imaging.** All imaging was conducted using a dual wide-field and 2-photon in vivo microscope (Neurolabware LLC). Widefield calcium imaging was performed with blue excitation light (479 nm LED, 469/35 nm bandpass filter, Thorlabs) through a low magnification (2x) objective lens (NA = 0.055, Mitutoyo) and green emission light (498 nm dichroic, 525/39 nm filter, Semrock) collected with an EMCCD camera (QImaging EM-C2, Teledyne Photometrics; 1004 by 1002 pixels with 2 by 2 binning) at a rate of 5 frames per second (fps). For 2-photon imaging, a 920-nm femtosecond laser beam (Newport MaiTai DeepSee) was raster-scanned using galvo and resonant scanners (Cambridge 6215H and CRS 8K) and focused at 100–300 nm depth below the pial surface using a 16x lens (NA = 0.8, Nikon). GCaMP6 fluorescence was collected using a bandpass filter (510/84 nm, Semrock) and GaAsP hybrid photodetectors (H11706-40, Hamamatsu), and images were reconstructed and acquired using Scanbox (version 4.0, https://scanbox.org). Individual imaging planes (720 x 512 pixel per frame, 1249 by 1067 um field-of-view) were collected at different depths. Simultaneous multi-plane imaging was achieved by rapidly changing the focus with an electrically tunable lens (EL-10-30-TC, Optotune AG; staircase mode) from 100 to 300 um depth and varying the laser power (from ~50 to 120 mW). Images were acquired from 3 or 4 evenly spaced planes at rates of 10.33 and 7.75 Hz, respectively. Blackout material (Thorlabs) blocked stray light from the visual display entering the collection light path.

**Data analysis.** All data analysis was performed using custom scripts written in Python or MATLAB
(The Mathworks, Natick, MA).

**Retinotopic mapping and area delineation.** Retinotopic maps were quantified by averaging the acquired camera images across repeated visual stimulation trials (Supplementary Fig. 1a, right) and calculating the phase angle of the visual stimulus location evoking maximal calcium responses (Supplementary Fig. 1a, left). Because each area has a near-complete visual representation, the trajectory of the visual stimulus results in multiple circular activation patterns, and pinwheel-like clockwise and counter-clockwise phase maps (Supplementary Fig. 1a, right). The border between V1 and the surrounding HVAs was delineated by tracing the gaps around the main activation pattern in the posterior cortex. The borders between HVAs were drawn at the reversal of the phase map directions. Using published delineations[19,20], a template was applied on the cortical surface with varying scale, rotation, and translation (Supplementary Fig. 1b).

**Calcium imaging data.** To correct x-y motion, 2P images for all experiments during the imaging sessions were registered to a common reference image (by registering and averaging 1200 frames from the center of the session). Regions of interest (ROIs) of active neural cell bodies were detected by computing local correlation (3 by 3 pixel neighborhood, threshold at correlation coefficients >0.9; customized MATLAB software) and identifying spatially connected pixels, selecting ROIs with near-circular shapes (maximum length/width aspect ratio <2). Cellular fluorescence time courses were generated by averaging all pixels in an ROI, followed by subtracting the averaged neuropil signals in the surrounding ring (morphological dilation, ring size matched to the ROI, nonoverlapping with neighboring ROIs) and correcting for slow baseline drift. Raw calcium time courses ($\Delta F/F_0$) were expressed as fractional changes to the baseline fluorescence.

**Selection of visually-responsive cells.** A cell was classified as responsive if the median time courses computed across trials showed a response peak with magnitude >3x standard deviation of the pre-stimulus activity (averaged >2 s) for over a continuous period > 1 s for at least one stimulus condition. Cells responding to the visual stimulus offset but not during visual stimulus epochs were excluded from the analysis. Cells were selected based on a response reliability index, $r$, defined as the 75th percentile of the cross-trial correlation coefficients of the de-randomized response time courses ($r > 0.3$, Supplementary Fig. 3a, example responses). Cells with $r > 0.3$ have high amplitude responses (Fig. 1h) and significantly higher reliability than shuffled responses (Supplementary Fig. 3c; 1000x randomization, reporting values at 97.5 and 99.75 percentiles). Neurons' response amplitudes were calculated as the average $\Delta F/F_0$ over the duration of the stimulus epochs.

**Tuning analysis**
*Spatiotemporal tuning.* Responses were described by two-dimensional elliptical Gaussian functions[12,31]:

$$R(sf, tf) = A\exp\left(-\frac{(\log_2 sf - \log_2 sf_0)^2}{2(\sigma_{sf})^2}\right)\exp\left(-\frac{(\log_2 tf - g(sf))^2}{2(\sigma_{tf})^2}\right) \quad (1)$$

where $A$ is the peak response amplitude, $sf_0$ and $tf_0$ are the peak spatial and temporal frequencies, $\sigma_{sf}$ and $\sigma_{tf}$ are the spatial and temporal frequency tuning

widths, and

$$g(sf) = \xi\left(\log_2 sf - \log_2 sf_0\right) + \log_2 tf_0 \quad (2)$$

In this expression, $\xi$ is the speed tuning index (*SI*) that captures the slant in the spatiotemporal frequency space. Model parameters were estimated by searching for the model fit with least squared errors in respect to the raw responses (MATLAB *lsqcurvefit* function). Fit quality was quantified as the normalized root mean squared (RMS) error between measured and model responses as a fraction of the peak magnitude of responses. Fits with a normalized RMS error <0.1 (10%) were selected for quantification (see Table 1 for cell numbers). The low and high cutoffs for the spatial and temporal frequency (cutoff frequency at half-maximum amplitude) were measured by evaluating the functions $R(sf, tf_0)$ and $R(sf_0, tf)$, respectively. A cell was defined as: lowpass if the low cutoff frequency is lower than the lowest frequency tested (i.e., 0.02 cpd or 0.5 Hz), highpass if the high cutoff frequency is higher than the highest frequencies tested (i.e., 0.32 cpd or 16 Hz), and bandpass if the low and high cutoffs are within the bounds of the spatial and temporal frequencies tested.

*Tuning and preference for stimulus anisotropy.* The preference for anisotropic versus isotropic stimuli was quantified with an ANISO/ISO preference index (*API*):

$$API = \frac{R_1 - R_0}{R_1 + R_0} \quad (3)$$

where $R_1$ and $R_0$ are the peak amplitudes of responses to anisotropic (15 deg FWHM orientation bandwidth) and isotropic stimuli (infinite orientation bandwidth), respectively. For Fig. 5b, the responses to ISO and ANISO stimuli were used (dataset 1 and 2). In Fig. 6c, the responses to the isotropic and the most elongated stimuli (15 deg FWHM) at the neurons' preferred spatial frequencies were used (dataset 3). The tuning curves for stimulus anisotropy (elongation) were quantified as 4-element vectors of trial-averaged response amplitudes to four orientation bandwidth at the neuron's preferred spatial frequency (dataset 3).

*Orientation tuning.* Selectivity for orientation was quantified from the time course of responses to anisotropic stimuli at the neurons' peak spatiotemporal frequencies using the temporal evolution of calcium signals as an approximation of the responses to varying orientation. Because neurons tuned to 0-deg orientation show calcium activity after both the onset and the offset of the visual stimulation, calcium responses after the visual stimulus offset were wrapped around (i.e., they were added to the onset responses), providing a more accurate estimate of responses as a function of stimulus orientation. The resulting time courses were binned and used for orientation tuning analysis. The orientation selectivity index (*OSI*) is approximated with the equation $OSI = 1\text{-}CV$[58]. The circular variance is defined as

$$CV = 1 - \left|\frac{\sum_k r_k e^{i2\theta_k}}{\sum_k r_k}\right| \quad (4)$$

where $r_k$ is the average amplitude of responses at orientation angles $\theta_k$ (from 0 to 180 deg in radian units).

Simulations were used to validate the above method for quantification of orientation tuning by comparing *OSI* estimates from spike ($OSI^{spike}$) and calcium activity ($OSI^{calc}$) (Supplementary Fig. 11). Gaussian tuning curves and Poisson spiking neurons with zero spontaneous firing rates were used to generate time-varying spike trains simulating responses to varying orientation. The spike trains were convolved to obtain simulated calcium responses. The spike trains were binned by orientation and $OSI^{spike}$ was calculated. The convolved calcium traces were obtained by applying a convolution kernel to the model spike trains (Supplementary Fig. 11b). The convolution kernel was a sum of two exponentials with typical GCaMP6s transient parameters (rise time constant: 200 ms; decay time constant: 560 ms)[43,70]. $OSI^{calc}$ was obtained using the abovementioned procedure (last paragraph). We repeated this analysis on model cells with different preferred orientations and orientation bandwidth (10 to 170 deg FWHM) (see example responses in Supplementary Fig. 11d) and examined the relationship between the resulting $OSI^{spike}$ and $OSI^{calc}$ values (Supplementary Fig. 11e).

*Decoding analysis.* We decoded neuronal population activity using linear support vector machine (SVM) classifiers[67,71] to test how well the classifiers discriminate response patterns for stimulus pairs with different or identical motion speeds (i.e., speed pairs and iso-speed pairs) using dataset 1. The speed pairs are defined as neighboring stimulus pairs lying orthogonally to the iso-speed lines (Fig. 4a) with a modest (4-fold) difference in speed. Switching the combination of the spatial and temporal frequencies of speed pairs yields iso-speed pairs, which have identical speeds (Fig. 4d). SVM classifiers were trained and tested in pairwise classification for all possible pairs. Visual responses across four trials were split into training and testing groups (half-half). The decoding accuracy was defined as the proportion of correct classification decisions to the testing groups (standard cross-validation). A resampling procedure was used to equalize the pool size across areas. For each area, the decoding analysis was repeated 50 times with random resampling of neurons (with replacement). Each time included 100 iterations of training and testing with a random set of trials (without replacement). We calculated the average decoding performance across iterations for each sampled pool and the averages across the sampled pools to estimate the 95% confidence intervals of the performance. To test

the scaling of decoding performance as a function of pool size, we measured decoding performance as a function of the pool size (a logarithmic increase from 2 to 512 neurons, without replacement) per area. The decoding performance increases with pool size and eventually reach 100 percent accuracy for most stimulus pairs, but with different ascending slopes (Fig. 4b, e, insets). To quantitatively compare decoding performance between areas, we reported the decoding accuracy of classifiers with a small pool size (16 neurons; Supplementary Fig. 9a, c) and estimated the relative changes in the decoding accuracy for speed pairs in higher visual areas with respect to V1 (Fig. 4b, c), and between speed pairs and corresponding iso-speed pairs in each area (Fig. 4e–f). The means and 95% confidence intervals of the difference in decoding accuracy were used to determine the statistical difference levels, which were reported in Supplementary Fig. 9b, d.

*Hierarchical bootstrap.* For statistical analyses and clustering of data across visual areas, a hierarchical bootstrap[72] of multi-level data was used. Using a randomly resampling approach (with replacement), the responses of specific visual areas was represented by a hierarchical dataset of size N-M-L, where N is the number of cells per animal, M is the number of animals per resampling, and L is the number of resampling. Statistical and clustering analysis was then applied as described as follows.

*Statistical analysis of tuning parameters.* To estimate probability density functions of tuning parameters, we use kernel density estimators with a bandwidth of 0.05 or 0.1 (MATLAB *ksdensity* function). To assess the statistical significance of differences between the density functions, we used two-sample Kolmogorov-Smirnov tests applied to randomly generated hierarchical bootstrap samples (two-sided hierarchical bootstrap KS test). For each pair of areas, we compared hierarchical datasets using a two-sample KS test (MATLAB *kstest2* function). The test statistic $D$ was used to compute confidence intervals ($CI_{low}$ and $CI_{high}$, at $1/2*\alpha*100$ and $100-1/2*\alpha*100$ percentile, where $\alpha$ is the significance level), which were compared to critical values $D_\alpha$ corresponding to different significance levels ($\alpha = 0.05, 0.01,$ and $0.001$) where $D_\alpha$ is calculated as the following:

$$D_\alpha = c(\alpha)\sqrt{\frac{n_1+n_2}{n_1 n_2}} \qquad (5)$$

where $n1$ and $n2$ are the sample sizes of the two tested datasets, $c(\alpha)$ is a coefficient determined by $\alpha$ [$c(\alpha) = 1.36, 1.63, 1.95,$ for $\alpha = 0.05, 0.01$ and $0.001$, respectively]. If $CI_{low}$ is greater than any $D_\alpha$, the most significant level applicable was reported; otherwise, no significant difference between the two distributions was reported. See Supplementary Fig. 5a, b for a demonstration of these procedures.

*Clustering.* Responses to stimuli of different degrees of anisotropy (dataset 3) were clustered into three groups of distinct tuning curves for anisotropy (MATLAB *clusterdata* function, ward linkage approach). A hierarchical bootstrapping approach was used to generate ten sub-datasets per area; each is a matrix of the anisotropy tuning curves of 500 random reliable cells. All sub-datasets across areas were pooled, and the tuning curves were clustered into three groups: non-OS (non-orientation-selective), sharp-OS, and broad-OS. The fraction of these groups in each area was averaged across sub-datasets and reported in Fig. 6f.

Responses to spatiotemporal frequencies and orientation (dataset 1 and 2) were grouped using unsupervised clustering. A training dataset was constructed with random sampling (with replacement) of the activity of 2000 neurons per area responding to either ISO or ANISO stimuli (dataset 1 and 2). Each neuron is assigned a vector of responses to ISO and ANISO stimuli (60 stimulus conditions, response amplitude normalized to range from 0 to 1), followed by extracting the first 12 principal components of the training dataset (explaining ~89% of the variance in the data, the original response matrix was too large for the clustering algorithm). A squared-Euclidean distance matrix of the new dataset was calculated and then fed into different clustering algorithms.

To determine the clustering method and the number of clusters that yield the best separation between clusters, we used a silhouette analysis. The k-means clustering method outperformed spectral clustering and ward linkage clustering, yielding the highest overall silhouette coefficients (Supplementary Fig. 13c). Moreover, silhouette analysis revealed a maximal coefficient at around 12 clusters. These clustering results were supported by two other independent validation measurements with Davies-Bouldin and Calinski-Harabasz indices[73] (Supplementary Fig. 13d). Afterward, the centroid of each cluster was obtained, and neurons of the entire dataset were assigned to clusters with the nearest centroids.

To quantify the proportion of clusters within each area with the consideration of inter-animal variability, we used a hierarchical bootstrapping approach (1000 bootstraps, 150 neurons per mouse, 5 mice per area). From the distributions of bootstrapping results, the means and 95% confidence intervals of the proportions were obtained and expressed as the ratio to the chance level (1/12), where the 12 clusters were evenly presented. To determine whether specific clusters are over- or underrepresented in each area, we compared their confidence intervals to the chance level and determined whether the differences were significant. To quantify the similarity between areas in terms of the proportions of encompassed clusters, we calculated the means and standard errors of the mean of Pearson's correlation coefficients between areas using the hierarchical bootstrapping datasets (Fig. 8g).

To determine whether area X's similarity with area Y ($S_{xy}$) is significantly different from its similarity with area Z ($S_{xz}$), and to quantify the difference, we computed the standardized mean difference, or *Cohen's d*, as the following,

$$d = \frac{|M_{xy} - M_{xz}|}{\sqrt{(SD_{xy}^2 + SD_{xz}^2)/2}} \qquad (6)$$

where $M_{xy}$ and $M_{xz}$, $SD_{xy}$ and $SD_{xz}$ are the mean values and standard deviations of the corresponding bootstrap datasets $S_{xy}$ and $S_{xz}$. We defined the effect size of difference with *Cohen's d* < 0.2 as small, between 0.2 and 2 as median, and greater than 2 as large (Supplementary Fig. 14).

To visualize the composition of functional clusters in different areas, we used t-SNE to map neurons in a two-dimensional representation. The clustering training dataset (16,000 neurons x 12 principal components) was fed to the t-SNE algorithm (MATLAB *tsne* function) with a large perplexity value of 120 to better present the global structure of the dataset. The learning rate was set to $n/10$ as a recommended value for large datasets, where $n$ is the number of cells[74].

**Reporting summary.** Further information on research design is available in the Nature Research Reporting Summary linked to this article.

## Data availability
Source data are provided with this paper. Source data for Figs and Supplementary Figures are included in the Source Data files. Datasets 1–3 and cellular response time courses generated in this study are openly available on figshare (https://doi.org/10.6084/m9.figshare.18415925). Raw image data are too large (>4 TB tiff files) to be distributed. These files are archived and maintained in local servers in the host institute, and are openly available upon request to the corresponding author. Source data are provided with this paper.

## Code availability
Source code is available on figshare (https://doi.org/10.6084/m9.figshare.18415925).

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

## Acknowledgements

We are grateful to Joao Couto for establishing the imaging setup and support with image processing, Asli Ayaz, and Karl Farrow for feedback, discussions, and comments on earlier versions of the manuscript. This work was supported by Neuro-Electronics Research Flanders (V.B.), Research Foundation Flanders (FWO) fellowship 12E4314N (B.V.) and grant G0D0516N (V.B.), and KU Leuven Research Council grant C14/16/048 (V.B.).

## Author contributions

Conceptualization: X.H. and V.B.; Methodology: X.H., B.V., and V.B.; Software: X.H. and B.V.; Investigation: X.H.; Data curation: X.H.; Formal Analysis: X.H. and V.B.; Visualization: X.H. and V.B.; Writing—original draft: X.H.; Writing—review & editing: X.H., B.V., and V.B.; Funding acquisition: V.B.; Resources: X.H. and V.B.; Supervision: V.B.

## Competing interests

The authors declare no competing interests.
