## [Peer Review File · Nature Communications]

REVIEWER COMMENTS

Reviewer #1 (Remarks to the Author):

The manuscript systematically characterizes spatiotemporal frequency tuning of neurons across mouse primary visual cortex and higher visual areas. The key findings are:

- 1) Mouse V1 and higher visual areas differ in their selectivity for spatial and temporal frequencies.
- 2) Higher visual areas tend to have more speed tuned cells and speed is more readily decoded from their responses at certain spatiotemporal frequencies.
- 3) Higher visual areas and V1 show more subtle difference in their preferences for isotropic and anisotropic (oriented) stimuli, with the exception of area Li, which primarily contains isotropic-preferring cells.
- 4) Neuronal responses can be classified based on their preference for spatiotemporal frequencies and isotropic/anisotropic stimuli into ~12 functional classes. V1 and HVA differ in the abundance of these functional classes.

Many of the conclusions of the manuscript were known before or hinted at by prior work. However, as this dataset is more systematic and complete than prior studies, I believe it will serve as an excellent reference for future work examining functional specialization of mouse visual cortical areas. The comparison of isotropic and anisotropic stimuli is novel and interesting.

In brief, I think this is an excellent study albeit of somewhat specialist interest. My concerns are largely minor.

1. The color scale in Figure 4C and F is confusing and potentially misleading. The use of black for negative differences in decoding accuracy hides the fact that V1 outperforms HVAs for some SF/TF combinations. For example, a blue / red colormap with blue for negative values would be more intuitive and easier to read.

The corresponding text in the Results is also somewhat unclear and does not fully reflect the data (lines 259-263): "Consistent with their enriched visual speed representations, area LM, dorsal visual areas AL, RL, AM, and area LI show increased accuracies relative to V1 at specific spatiotemporal frequencies (Figure 4C; Figure S9C, statistical significance). Area PM, in contrast, did not show such an advantage over V1." Please revise to make explicit the fact that V1 in fact shows an advantage over HVAs for certain SFs/TFs.

Nevertheless, it is clear the anterior HVAs do perform better at speed compared to isospeed discriminations.

2. Please report the luminance and contrast of the stimuli.
3. The Methods (line 644) mention volume imaging but this is not mentioned anywhere else. Were the functional data acquired across multiple planes simultaneously and if so what was the resulting volume rate?
4. To estimate orientation tuning and account for the dynamics of GCaMP6s, "the residual calcium response after the stimulation offset was added to the onset response." Could this not potentially

inflate the estimate of responses to 0 degrees, and they might be double counted if present both immediately after the onset and offset of the stimulus? In addition, some neurons might respond to the offset of the stimulus itself, akin to suppressed by contrast cells in the retina, irrespective of stimulus orientation.

5. Since mice were free to run during the recording sessions, an analysis of any differences in modulation of neurons in V1 and HVAs by running speed would be interesting and would add to the completeness of the manuscript.

6. Would be great to see how the speed tuning index looks across response clusters and maps onto the t-SNE projection in Figure 8.

7. The use of “tunings” and “neural activities” is somewhat strange. In the field, “tuning” and “activity” are usually used as uncountable nouns and one would normally use them as singular nouns (“tuning” and “neural activity”) even when referring to responses of many neurons.

8. Line 707: typo – “werer”

Reviewer #2 (Remarks to the Author):

In this manuscript Han et al. do a very comprehensive survey of visual response properties to parameterized stimuli across the mouse visual cortex. The physiology is quite comprehensive, including >30k visually-responsive neurons imaged across identified visual regions. However, while the data set is impressive, I was less enthusiastic about the analysis and interpretation of the data, as discussed below.

Major Concerns:

My principal concern is that I found the key results of the manuscript either incremental or difficult to interpret. In the conclusion, the authors argue that their results yield four key insights. I'll discuss each in turn:

1) The first key result is that each visual cortical area has a unique representation of spatial and temporal features. Their paper unequivocally shows this, but this has been known for about a decade (Andermann et al, 2011; Marshall et al., 2011). It is true that the current study is much more thorough than the previous studies (with orders of magnitude greater recorded neurons), but the results are largely overlapping with prior established findings.

2) The second key result is that the diversity of responses are explained by ~12 distinct channels, each of which contains neurons with distinct response properties. I had a lot of trouble understanding this point, as I believe the number of “channels” necessary to explain the data very much depends on the stimulus parameters used for the clustering. For example, if they also included responses to texture, coherent motion, or other stimulus features, they likely would have found additional channels.

3) The third key result is that they find a profound anterior-to-posterior gradient in the spatiotemporal response properties of the individual neurons. This is an interesting hypothesis, and there is some support for this from recent papers. However, I found the evidence for this argument to be lacking. For

reasons I was not totally clear on, they only sampled V1 along the horizontal meridian, so they did not have anatomical data on gradients of response properties within V1 (and most of the other regions were too small to get an appreciable anterior-to-posterior mapping). Instead, as far as I could tell, the argument hinged on the fact that the three more anterior regions had different response properties than the three more posterior regions. I did not find this a compelling argument for a strong cortex-wide gradient of response properties.

4) The last key result is that there are a number of neurons that respond to non-oriented stimuli, and that non-oriented features are also encoded in visual cortex. Similar to the first result, they showed this very convincingly, but I am not sure how they went beyond what is currently known in the field (I think most vision researchers would find this self-evident).

Taken together, while the data set is quite comprehensive and impressive, and their analyses were sophisticated and technically sound (a few quibbles aside), I am not sure whether the manuscript advances our understanding of visual processing.

Minor:

1) The limited set of visual stimuli also leads to some counterintuitive results, such as their finding that PM is most similar in terms of response properties to lateral visual region LI, despite a number of papers finding anatomical divergence between the regions (Wang et al., J Neuro, 2011; Wang et al., J Neuro, 2012) and that the regions respond very differently to gratings and RDK stimuli (e.g., Smith et al., Nat Neuro, 2017). This should be discussed.

2) On a related note, their findings on PM seem to contradict previous findings (Andermann et al., 2011) that found considerable speed tuning in PM (for slower speeds). The data are the data, but the discrepancy should be discussed.

3) The authors often use speed tuning interchangeably with motion processing. However, although speed tuning is a component of motion processing, most researchers consider motion processing to also encompass detection of coherent motion (measured using random dot kinematograms or plaid stimuli not used in the study). Indeed, the ventral stream regions that exhibit the largest responses to coherent motion stimuli appeared not have the greatest speed tuning in this study. This should be clarified in the results and mentioned in the discussion.

4) They mention it briefly, but since they already have the data, the manuscript would benefit from a supplementary figure showing that eye movements, pupil dilation, and running speed do not affect the response properties of the neurons.

5) Instead of using an arbitrary cutoff ($CC > 0.3$) for responsive cells, it would be preferable to test the hypothesis that the CCs are greater than some null distribution (e.g., the CCs calculated from circularly shuffled responses).

6) The supplementary figure ordering jumps around a bit (e.g., Fig S7 appears in the text before Fig S4).

Other comments:

- 1) I'd like to commend the authors on using appropriate statistics (such as hierarchical models) to avoid independence of sample issues resulting from population measurements within the same mice.
- 2) If the manuscript is accepted, I would strongly encourage the authors to archive the processed 2P data in a public repository so the scientific community can access the data.

Reviewer #3 (Remarks to the Author):

This is an impressive and welcome study of the distribution of spatiotemporal response properties across areas of mouse visual cortex. Using rich noise stimuli to drive calcium responses in awake head-fixed mice, the results show that the parietal areas AL, RL and AM are sensitive to fast moving objects, and show strong orientation selectivity to contours at low spatial resolution. In contrast, the medial area, PM, and lateral area, LI, prefer slow speeds, show high spatial acuity but differ in the high (PM) and low (LI) strength of orientation selectivity. Based on these findings the authors propose that visual information is processed in a dorsal stream specialized for motion processing and two additional branches representing slow motion at high spatial resolution and distinct shape sensitivities. How the proposed streams map onto dorsal and ventral streams observed in the patterns of intracortical connections remains an open question. The results suggests that cortical networks in which spatial features are encoded from motion instead of orientation cues may have fundamentally different organizations.

The strength of the study is the demonstration that 12 types of neurons, distributed in area-specific patterns underlying area-specific functions, encode spatiotemporal properties. Although the results strongly support area-specific specializations for processing shape and the speed of motion, the underlying substrate argues for highly distributed connections, which are only partially aligned with known streams. Extracting distinct channels from such highly overlapping flow patterns in the absence detailed structural analyses of seems premature.

The article refers to "channels" and "streams" interchangeably. Historically, channels were used to describe pathways, which carry low-level sensory cues that lead to a single aspect of perception. In contrast, streams refer to processing networks which carry several low level cues and give rise to several distinct attributes of objects Conversely attributes of percepts can be inferred from several sensory cues not just one. The article describes how the product of motion and shape cues are distributed across different areas of mouse visual cortex. This comes about, I would assume, by streams not channels. I therefore recommend to use in the title and throughout the text "streams" and avoid sentences such as "organization of visual channels and...specialized streams" (e.g. lines 23, 90).

Line 114. Visuotopic mapping of V1 shows that the cells displayed in Fig. 1C are unlikely at the horizontal meridian as indicated in the text. Rather the majority of cells are presumably located in the upper

temporal field. The problem of assigning the horizontal meridian at a location far too posterior in V1 emerges again in Fig. S7A, B, D. This should be corrected and the impact on the distribution on the tuning across visual space (Fig. S7C, E) should be assessed.

Line 115. The text implies that recordings along the horizontal meridian is advantageous for comparing responses in different areas. I would challenge this assessment. For example, the borders of areas V1, LM, AL and RL, and AM and PM are in close proximity of the horizontal meridian, which makes assigning cells to specific areas problematic. Because of this potential problem, I recommend re-analysis of data by using sliding windows parallel and perpendicular to the lateral and medial borders of V1. Then plotting the functional properties in bins and find significant changes in spatiotemporal response profiles. The analysis will make no specific assumption of the exact location of the areal borders as drawn in Fig. 1C, but would expect that the borders emerge as functional changes between areas. The recommended re-analysis includes all figures in which areas are compared.

Figure 1C shows cells on the posterior side of LM and LI, located in the territory of P and POR. It is not clear whether these cells are included in the analysis. If not this should be stated in the text.

How suited are Thy1-GCaMP6s mice for widefield imaging? Is there any concern that the pattern of expression selects for specific cell types? It would be good to include short comment.

Line 32. It is misleading to suggest that “neurons in the visual cortex form parallel channels”. While parallel channels exist in the geniculocortical pathway, evidence indicate that the channels are mixed in the cortex. The same principle applies to visual cortex of monkey, cat and mice.

Line 130. The text states that stimuli were presented -30~50 deg elevation. From the plot in Fig. 1C, recordings in the far lower field seem very unlikely. Also see Line 616.

Line 150. The question is whether responses differ across areas. Whether the expected functional differences are aligned with specific channels is an interpretation that can only be derived in conjunction with intracortical connectivity and does not directly result from the recordings.

Line 161. “Profound specificity” is hyperbole and should be deleted.

Line 179. Consistently observed is sufficient. Delete “robustly”.

Line 204. “...share visual channels.” Replace channels by “attributes”.

Line 215. Figure 3C, inset?

Line 315. Replace the title by “Areal preferences for orientation”.

In a paper about processing channels, the misalignment of the functional properties recorded in AL and PM with the proposed association with the dorsal stream is a prominent topic that requires discussion that is more detailed.

Line 488. The sentence “The integration...” needs editing.

Line 497. Davide Zoccolan has shown in rats that ventral areas play a role in object recognition.

Line 614. Please indicate the size of stimuli. Full field?

RESPONSE TO REFEREE LETTER

We thank the reviewers for their thorough assessment and insightful comments. Following their comments, we improved the analyses and added controls to address the issues identified. We also revised the manuscript thoroughly to describe the findings more clearly.

This revised manuscript provides a sharper focus on the functional organization of processing streams. It features complete re-analysis of the data including a revised areal parcellation. We now present separate data for area POR/P as well as a new mesoscopic analysis of the neurons' spatiotemporal tuning properties showing consistent shifts in representations at retinotopic borders. Finally, we added a simulation showing that selectivity for orientation can be faithfully estimated from the time course of calcium responses (Figure supplement 11).

The reviewers acknowledge that the study represents one of the most comprehensive and better-designed studies in the field. A reviewer stated the paper could become a landmark reference in the field. With regards to advances made, we note that the study expands the evidence for specialized spatiotemporal representations in the mouse from three to eight visual areas (see response to Reviewer 2) and provides much needed functional evidence for organized dorsal and ventral visual streams in the mouse cortex. Thirdly, the study provides experimental support for theoretical studies predicting that selectivity for both oriented and non-oriented features is important for faithful scene encoding (e.g. Olshausen et al., 2009), revealing a previously overlooked population that encodes non-oriented features. This is not a trivial finding. For decades, orientation tuning and edge detection has been considered as the hallmark of processing the visual cortex. The population we have uncovered has diametrically opposite preferences to what has been described in the mouse cortex literature. Moreover, the unique enrichment of these neurons in ventral areas provides strong functional evidence for dorsal-ventral segregation in the mouse.

Finally, the study presents promising avenues to tackle functional diversity in the visual cortex, addressing functional cell types and consequences for stimulus encoding. Though we found a spectrum of properties in terms of spatiotemporal tuning, we uncovered a clear trichotomy in spatial integration properties and new evidence of specializations in HVAs. This approach could be expanded in the future to include a broader set of stimulus dimensions.

We present below our point-by-point response to the comments.

Reviewer #1 (Remarks to the Author):

The manuscript systematically characterizes spatiotemporal frequency tuning of neurons across mouse primary visual cortex and higher visual areas. The key findings are:

1) Mouse V1 and higher visual areas differ in their selectivity for spatial and temporal frequencies.

2) Higher visual areas tend to have more speed tuned cells and speed is more readily decoded from their responses at certain spatiotemporal frequencies.

3) Higher visual areas and V1 show more subtle difference in their preferences for isotropic and anisotropic (oriented) stimuli, with the exception of area Li, which primarily contains isotropic-preferring cells.

4) Neuronal responses can be classified based on their preference for spatiotemporal frequencies and isotropic/anisotropic stimuli into ~12 functional classes. V1 and HVA differ in the abundance of these functional classes.

Many of the conclusions of the manuscript were known before or hinted at by prior work. However, as this dataset is more systematic and complete than prior studies, I believe it will serve as an excellent reference for future work examining functional specialization of mouse visual cortical areas. The comparison of isotropic and anisotropic stimuli is novel and interesting.

In brief, I think this is an excellent study albeit of somewhat specialist interest. My concerns are largely minor.

As explained above, we have revised the manuscript to clarify advances over previous studies. We have also revised the text to improve accessibility to the journal's readership.

1. The color scale in Figure 4C and F is confusing and potentially misleading. The use of black for negative differences in decoding accuracy hides the fact that V1 outperforms HVAs for some SF/TF combinations. For example, a blue / red colormap with blue for negative values would be more intuitive and easier to read.

Thank you for pointing this out. We have corrected the color map.

The corresponding text in the Results is also somewhat unclear and does not fully reflect the data (lines 259-263): "Consistent with their enriched visual speed representations, area LM, dorsal visual areas AL, RL, AM, and area LI show increased accuracies relative to V1 at specific spatiotemporal frequencies (Figure 4C; Figure S9C, statistical significance). Area PM, in contrast, did not show such an advantage over V1." Please revise to make explicit the fact that V1 in fact shows an advantage over HVAs for certain SFs/TFs.

We added:

Decoding pairs of stimuli of nearby speeds, we found V1 and HVAs show distinct decoding performance patterns with area-specific dependence on spatiotemporal frequency (Fig. 4c, Supplementary Figure 8a). While the V1 show moderate decoding performance across the frequency spectrum with a maximal accuracy around 75%, many HVAs present more biased decoding performances, showing near-perfect accuracy at specific regions (around slopes of average tuning maps) and chance-level accuracy (around the null frequency and the peak frequency) (Fig. 4c, Supplementary Figure 8b).

Nevertheless, it is clear the anterior HVAs do perform better at speed compared to isospeed discriminations.

2. *Please report the luminance and contrast of the stimuli.*

This is now described in Methods:

The stimuli had 50% standard root-mean-square contrast, 80 cd/m² mean luminance at the center of the screen with a gradual decrease to 54 cd/m² at the borders, due to the intrinsic property of the display.

3. *The Methods (line 644) mention volume imaging but this is not mentioned anywhere else. Were the functional data acquired across multiple planes simultaneously and if so what was the resulting volume rate?*

This is now described in Methods:

Volume imaging was achieved using a focus tunable lens (EL-10-30-TC, Optotune; staircase mode). We recorded activity from neurons typically between 100 to 300 μm deep below pia with evenly spaced 3 or 4 planes, resulting in sampling rates of 10.33 and 7.75 Hz for each plane respectively.

4. *To estimate orientation tuning and account for the dynamics of GCaMP6s, “the residual calcium response after the stimulation offset was added to the onset response.” Could this not potentially inflate the estimate of responses to 0 degrees, and they might be double counted if present both immediately after the onset and offset of the stimulus?*

We provide a new control analysis (Supplementary Fig. 11, Methods: orientation tuning analysis) showing the close relationship between the true OSI measured on simulated spike trains and the OSI measured on simulated calcium traces. The current method used to estimate OSI based on calcium traces slightly underestimates OSI for the most tuned cells. We also didn't observe biases at specific orientations using simulated data.

In addition, some neurons might respond to the offset of the stimulus itself, akin to suppressed by contrast cells in the retina, irrespective of stimulus orientation.

We apologize that this selection step was not stated in the original submission and now add it in the Methods.

Cells that showed responses to the offset of stimuli but no response during stimulus epochs were excluded from further analysis.

5. *Since mice were free to run during the recording sessions, an analysis of any differences in modulation of neurons in V1 and HVAs by running speed would be interesting and would add to the completeness of the manuscript.*

The treadmill was for the animal's comfort and the animals were neither trained nor incentivized to run. The running epochs are too few for a separate analysis and excluding the data from running epochs did not change the results.

6. *Would be great to see how the speed tuning index looks across response clusters and maps onto the t-SNE projection in Figure 8.*

We provide these plots in the new Fig. 7f and 8d, showing distinct functional cell groups have distinct speed-tuning properties.

7. *The use of "tunings" and "neural activities" is somewhat strange. In the field, "tuning" and "activity" are usually used as uncountable nouns and one would normally use them as singular nouns ("tuning" and "neural activity") even when referring to responses of many neurons.*

We corrected these throughout.

8. *Line 707: typo – "werer"*

We corrected the text.

Reviewer #2 (Remarks to the Author):

In this manuscript Han et al. do a very comprehensive survey of visual response properties to parameterized stimuli across the mouse visual cortex. The physiology is quite comprehensive, including >30k visually-responsive neurons imaged across identified visual regions. However, while the data set is impressive, I was less enthusiastic about the analysis and interpretation of the data, as discussed below.

Major Concerns:

My principal concern is that I found the key results of the manuscript either incremental or difficult to interpret.

In the conclusion, the authors argue that their results yield four key insights.

Reading through the comments below, we realized that the individual points below might have been understood as summaries of individual sections of the paper. This was not the intention. Rather, each point was meant as a summary of the whole paper. We hope the revised summary in the Discussion is clearer.

I'll discuss each in turn:

1) The first key result is that each visual cortical area has a unique representation of spatial and temporal features. Their paper unequivocally shows this, but this has been known for about a decade (Andermann et al, 2011; Marshel et al., 2011). It is true that the current study is much more thorough than the previous studies (with orders of magnitude greater recorded neurons), but the results are largely overlapping with prior established findings.

The high neuron count allows investigations that were simply not possible in previous studies including quantification of tuning diversity, identification of response types, examination of visual information streams, etc.

Furthermore, as explained in the Discussion, the study expands existing evidence of specialized spatiotemporal representations from three to eight cortical areas.

# cells			V1	PM	AL	LM	RL	AM	LI	POR/P
Marshel et al., 2011	Anesthetize	TF tuning	586	50	257	171	96	12	23	
		SF tuning	728	147	330	300	201	63	42	
Roth et al., 2012		SF/TF	399	192						
de Vries et al., 2020 (Emx1-layer 2/3)	Awake	TF tuning	341	104	180	453	100	49		
		SF tuning	209	58	117	208	31	19		
Andermann et al., 2011		SF/TF	87	46	107					
The current study		SF/TF	8718	2096	2609	6593	1656	1348	1216	2423

Table for reviewers. Comparison to previous studies.

The most complete dataset before this study, De Vries (2020), did not lay out tuning properties across areas. While AL and PM have been studied with a particular focus, this is not the case for other visual areas. The only comparable study covering multiple areas was Marshel et al. (2011), in which suboptimal visual stimuli and anesthesia were used, and the results were heavily biased (Supplementary Figure 6).

From Supplementary Figure 6; Comparison to Andermann 2011 and Marshel 2011.

2) The second key result is that the diversity of responses are explained by ~12 distinct channels, each of which contains neurons with distinct response properties. I had a lot of trouble

understanding this point, as I believe the number of “channels” necessary to explain the data very much depends on the stimulus parameters used for the clustering. For example, if they also included responses to texture, coherent motion, or other stimulus features, they likely would have found additional channels.

The underlying question is whether cortical neurons can be categorized into functional cell types based on their visual tuning. We agree that the conclusions about visual channels went too far given the data presented (cf Reviewer 3). We have therefore revised the manuscript to highlight how the analysis reveals areas’ multidimensional tuning properties, and how it reveals visual information streams and substreams. We also revised the text to state that the spatiotemporal tuning forms a wide spectrum rather than discrete clusters based on the clustering results.

3) The third key result is that they find a profound anterior-to-posterior gradient in the spatiotemporal response properties of the individual neurons. This is an interesting hypothesis, and there is some support for this from recent papers. However, I found the evidence for this argument to be lacking. For reasons I was not totally clear on, they only sampled V1 along the horizontal meridian, so they did not have anatomical data on gradients of response properties within V1 (and most of the other regions were too small to get an appreciable anterior-to-posterior mapping). Instead, as far as I could tell, the argument hinged on the fact that the three more anterior regions had different response properties than the three more posterior regions. I did not find this a compelling argument for a strong cortex-wide gradient of response properties.

This language was in reference to the differences in tuning between visual cortical areas, not a statement about the topographic organization of spatiotemporal preference. Although we did not explore response properties at different retinotopic locations, we added a new analysis showing the average spatial and temporal preferences in relation to the neuron’s location in the cortical surface (new Figure 3). The new analysis shows a clear correspondence between functional borders and retinotopic borders, and a relative uniformity of mean preferences within individual HVAs. We have revised the text to more clearly describe the differences across visual areas without implying topographic mapping.

4) The last key result is that there are a number of neurons that respond to non-oriented stimuli, and that non-oriented features are also encoded in visual cortex. Similar to the first result, they showed this very convincingly, but I am not sure how they went beyond what is currently known in the field (I think most vision researchers would find this self-evident).

This is not a trivial finding. For decades, orientation tuning and edge detection has been considered as the hallmark of processing the visual cortex. Our study explores regions of the visual stimulus space that have seldom been examined, finds a clear trichotomy of response patterns classifying neurons into non-overlapping groups, and uncovers a new response type that characterizes between 30 and 60% of mouse L2/3 cells. The finding of neurons that prefer non-oriented noise provides experimental support for theoretical studies predicting that selectivity for both oriented and non-oriented features is important for faithful scene encoding (e.g. Olshausen et al 2009).

Fig. 6d, a specific type of cell responding solely to non-oriented spatial components.

Equally importantly, our data revealed a unique enrichment of cells tuned to non-oriented features in the ventral but not dorsal areas, providing, arguably, the strongest functional evidence for parallel streams in the mouse visual cortex and open new venues for studying shape or scene processing using the mouse model.

Taken together, while the data set is quite comprehensive and impressive, and their analyses were sophisticated and technically sound (a few quibbles aside), I am not sure whether the manuscript advances our understanding of visual processing.

The revision puts a sharper focus on the functional organization of processing streams. We have revised it thoroughly to clarify that the study provides much needed functional evidence for parallel processing streams in the mouse visual cortex, and may even reflect a separate third visual information stream. This functional evidence strengthens existing anatomical evidence, from which testable hypotheses can be built to interrogate the wiring and computational rules in the mouse visual cortex as a model for mammalian vision.

Minor:

1) The limited set of visual stimuli also leads to some counterintuitive results, such as their finding that PM is most similar in terms of response properties to lateral visual region LI, despite a number of papers finding anatomical divergence between the regions (Wang et al., J Neuro, 2011; Wang et al., J Neuro, 2012) and that the regions respond very differently to gratings and RDK stimuli (e.g., Smith et al., Nat Neuro, 2017). This should be discussed.

The reviewer's comment pointed at one of the interesting observations of this study that area PM differs greatly from other dorsal areas but is highly similar to ventral areas, which is very much driven by their similar spatiotemporal preferences. Nevertheless, we also demonstrate PM clearly distinguishes from ventral areas in terms of the encoding for non-orientation components (Fig. 5-6). Given these pieces of functional and anatomical evidence from our and others' studies, which are also referred to by the reviewer, we built our hypothesis that PM might form a substream or a stream that runs in parallel to other dorsal areas and ventral areas. These are also discussed in the new discussion.

2) *On a related note, their findings on PM seem to contradict previous findings (Andermann et al., 2011) that found considerable speed tuning in PM (for slower speeds). The data are the data, but the discrepancy should be discussed.*

The discussion germane was provided in Lines 563-567 in the original submission. The relevant text in the new discussion is as follows:

While the diversity of spatiotemporal preferences we observe in AL and PM are generally consistent with results from studies in awake animals (Andermann et al., 2011; Glickfeld et al., 2013; de Vries et al., 2020) (Supplementary Figure 6), we observe weaker speed tuning in PM in comparison to AL, which seems to contradict previous studies (Andermann et al., 2011; Roth et al., 2012). The differences could reflect our use of noise stimuli instead of drifting gratings to measure speed selectivity. The stimuli differ in degree of motion coherence which could influence estimates of speed selectivity (Sit and Goard, 2020).

3) *The authors often use speed tuning interchangeably with motion processing. However, although speed tuning is a component of motion processing, most researchers consider motion processing to also encompass detection of coherent motion (measured using random dot kinematograms or plaid stimuli not used in the study). Indeed, the ventral stream regions that exhibit the largest responses to coherent motion stimuli appeared not have the greatest speed tuning in this study. This should be clarified in the results and mentioned in the discussion.*

The manuscript has been revised accordingly.

4) *They mention it briefly, but since they already have the data, the manuscript would benefit from a supplementary figure showing that eye movements, pupil dilation, and running speed do not affect the response properties of the neurons.*

We do not have enough repeated visual stimulation trials for this type of analysis. Also the treadmill was for comfort. Animals were not trained or rewarded for running. There are too few running epochs to allow for analysis.

5) *Instead of using an arbitrary cutoff ($CC > 0.3$) for responsive cells, it would be preferable to test the hypothesis that the CCs are greater than some null distribution (e.g., the CCs calculated from circularly shuffled responses).*

We chose a cutoff of trial-to-trial $CC > 0.3$ to exclude weak responders (Fig. 1h) from which quality tuning measurements could not be obtained. This cutoff corresponds approximately to 95th percentile of CC_{shuffle} , which is derived from shuffled datasets (new Supplementary Fig. 3b). We did look at the effect of varying cutoff on peak SF and TF preference (Supplementary Fig. 6a). As expected, decreasing cutoff decreases the separation between visual areas in terms of SF and TF preferences.

6) The supplementary figure ordering jumps around a bit (e.g., Fig S7 appears in the text before Fig S4).

The supplementary figures were reordered according to their order in the text.

Other comments:

1) I'd like to commend the authors on using appropriate statistics (such as hierarchical models) to avoid independence of sample issues resulting from population measurements within the same mice.

Hierarchical data analysis was already applied wherever possible in the original submission.

2) If the manuscript is accepted, I would strongly encourage the authors to archive the processed 2P data in a public repository so the scientific community can access the data.

Data and code to generate figures will be deposited here (<https://figshare.com/account/home#/projects/124309>).

Reviewer #3 (Remarks to the Author):

This is an impressive and welcome study of the distribution of spatiotemporal response properties across areas of mouse visual cortex. Using rich noise stimuli to drive calcium responses in awake head-fixed mice, the results show that the parietal areas AL, RL and AM are sensitive to fast moving objects, and show strong orientation selectivity to contours at low spatial resolution. In contrast, the medial area, PM, and lateral area, LI, prefer slow speeds, show high spatial acuity but differ in the high (PM) and low (LI) strength of orientation selectivity.

Based on these findings the authors propose that visual information is processed in a dorsal stream specialized for motion processing and two additional branches representing slow motion at high spatial resolution and distinct shape sensitivities. How the proposed streams map onto dorsal and ventral streams observed in the patterns of intracortical connections remains an open question. The results suggests that cortical networks in which spatial features are encoded from motion instead of orientation cues may have fundamentally different organizations.

The strength of the study is the demonstration that 12 types of neurons, distributed in area-specific patterns underlying area-specific functions, encode spatiotemporal properties.

Although the results strongly support area-specific specializations for processing shape and the speed of motion, the underlying substrate argues for highly distributed connections, which are only partially aligned with known streams. Extracting distinct channels from such highly overlapping flow patterns in the absence detailed structural analyses of seems premature.

We understand that visual channels may be interpreted as neural pathways, which is not the topic of this study. We have therefore revised the manuscript such that the possibility of visual channels is primarily represented as a possible interpretation of data.

The article refers to “channels” and “streams” interchangeably.

We believe this was one error in the introduction. We fixed it.

Historically, channels were used to describe pathways, which carry low-level sensory cues that lead to a single aspect of perception. In contrast, streams refer to processing networks which carry several low level cues and give rise to several distinct attributes of objects. Conversely attributes of percepts can be inferred from several sensory cues not just one. The article describes how the product of motion and shape cues are distributed across different areas of mouse visual cortex. This comes about, I would assume, by streams not channels.

We agree with these definitions, and have corrected the manuscript accordingly.

I therefore recommend to use in the title and throughout the text “streams” and avoid sentences such as “organization of visual channels and...specialized streams” (e.g. lines 23, 90).

We agree and have revised the manuscript accordingly.

Line 114. Visuotopic mapping of V1 shows that the cells displayed in Fig. 1C are unlikely at the horizontal meridian as indicated in the text. Rather the majority of cells are presumably located in the upper temporal field. The problem of assigning the horizontal meridian at a location far too posterior in V1 emerges again in Fig. S7A, B, D. This should be corrected and the impact on the distribution on the tuning across visual space (Fig. S7C, E) should be assessed.

This was an issue of writing and has been revised. The FOVs in higher visual areas were positioned to avoid V1. The FOVs in V1 were positioned to target the center of V1 which corresponds approximately to the center of the visual display (Fig. 1b).

Line 115. The text implies that recordings along the horizontal meridian is advantageous for comparing responses in different areas.

This text clearly was confusing. We did not mean to imply that.

I would challenge this assessment. For example, the borders of areas V1, LM, AL and RL, and AM and PM are in close proximity of the horizontal meridian, which makes assigning cells to specific areas problematic. Because of this potential problem, I recommend re-analysis of data by using sliding windows parallel and perpendicular to the lateral and medial borders of V1. Then plotting the functional properties in bins and find significant changes in spatiotemporal response profiles. The analysis will make no specific assumption of the exact location of the areal borders as drawn in Fig. 1C, but would expect that the borders emerge as functional

changes between areas. The recommended re-analysis includes all figures in which areas are compared.

Following the reviewer's suggestion, we provide clearer information about how visual areas were delineated (Figure 1a-c, Supplementary Fig. 1) and how FOVs were positioned. We also added a quantitative analysis of the neurons' SF/TF preferences and speed tuning in relation to the estimated retinotopic border. The analysis shows a close correspondence between functional properties and independently estimated retinotopic borders (new Fig. 3).

Figure 1C shows cells on the posterior side of LM and LI, located in the territory of P and POR. It is not clear whether these cells are included in the analysis. If not this should be stated in the text.

Those cells were not included in the analysis. In the revised manuscript, we assigned those cells to area POR/P and performed a thorough analysis.

How suited are Thy1-GCaMP6s mice for widefield imaging? Is there any concern that the pattern of expression selects for specific cell types? It would be good to include short comment.

These mice show strong uniform GCaMP6s expression and imaging their cortex produces high quality retinotopic maps (new Supplementary Figure 1). We demonstrated the signal in Supplementary Movie 1.

Line 32. It is misleading to suggest that "neurons in the visual cortex form parallel channels". While parallel channels exist in the geniculocortical pathway, evidence indicate that the channels are mixed in the cortex. The same principle applies to visual cortex of monkey, cat and mice.

Following the above suggestion, we have revised the introduction to focus on visual cortical processing streams.

Line 130. The text states that stimuli were presented -30~50 deg elevation. From the plot in Fig. 1C, recordings in the far lower field seem very unlikely. Also see Line 616.

The visual stimuli were spatially uniform random noise covering the entire visual display which covered 0 to 100 deg AZ and -30 to 50 deg EL. The V1 2P-FOVs are positioned to approximately cover the retinotopic representation of the center of the display.

Line 150. The question is whether responses differ across areas. Whether the expected functional differences are aligned with specific channels is an interpretation that can only be derived in conjunction with intracortical connectivity and does not directly result from the recordings.

Yes we agree. This is already addressed in the response to reviewer 2.

| *Line 161. “Profound specificity” is hyperbole and should be deleted.*

We revised accordingly.

| *Line 179. Consistently observed is sufficient. Delete “robustly”.*

We revised accordingly.

| *Line 204. “...share visual channels.” Replace channels by “attributes”.*

We revised the paragraph.

| *Line 215. Figure 3C, inset?*

The text was misplaced and it is gone.

| *Line 315. Replace the title by “Areal preferences for orientation”.*

We agree and have revised accordingly.

In a paper about processing channels, the misalignment of the functional properties recorded in AL and PM with the proposed association with the dorsal stream is a prominent topic that requires discussion that is more detailed.

This is addressed in the reply to reviewer 2. We also expand the discussion in the revision.

| *Line 488. The sentence “The integration...” needs editing.*

We revised accordingly.

| *Line 497. Davide Zoccolan has shown in rats that ventral areas play a role in object recognition.*

We are aware of relevant literatures, however believe the evidence for a role of the rat ventral areas in object recognition is still indefinite. We revised the text as the following.

Other higher-order ventral-like functions (i.e. object recognition) in the mouse or the rat, albeit reportedly existing (Tafazoli et al., 2017; Vermaercke et al., 2014), might not be as developed as those in higher visual mammals.

| *Line 614. Please indicate the size of stimuli. Full field?*

Indeed, the visual stimuli is presented full-screen. We add this information in the revised text.

REVIEWERS' COMMENTS

Reviewer #1 (Remarks to the Author):

As stated in my original review, I think this is an excellent paper that is bound to become a go-to reference for the field. Most of my earlier comments were related to clarifications of data presentation and methods and have been addressed.

The spatial maps of tuning properties in Figure 3 are a great addition. However, I do not think the presented data are sufficient to support the claims in lines 183-185: "Examining the distribution of average spatial and temporal tuning across the cortical surface, we observed sharp boundaries near the retinotopic borders, with biases that exceed the gradients observed within visual areas". Given the presented data, most features appear to vary smoothly across the cortical surface with the possible exception of discontinuity in TF tuning between areas PM and AM. It would be great to also include a spatial map of ISO/ANISO selectivity.

In the discussion, the sentence on lines 397-400 ends in a colon and it is unclear what the "three visual information streams" refers to. I am guessing that this is related to the discussion in lines 448-457 suggesting that area PM may be a part of a separate processing stream. Perhaps the discussion text could be restructured to clarify.

Minor:

1. Line 194: "which shows much less difference between areas" – needs to be rephrased, perhaps "which shows much fewer differences between areas"?

Reviewer #2 (Remarks to the Author):

The authors have significantly improved the analyses and discussion in the revised manuscript. In my view, they have done a nice job of addressing reviewer concerns and highlighting the scientific advance in their work. I have some (mostly minor) concerns listed below that should be addressed, but am otherwise in support of publication.

1) Line 21: "form a continuum" is an unusual phrasing. I would suggest "lie along a continuum".

2) Line 80: "propound" should be "profound".

3) Line 382: the clause "albeit nonuniformly" should have commas before and after.

4) Line 509-510: This is a declarative statement that has no data supporting it. It would be nice if the authors included an analysis of pupil dilation, eye movements and running in a supplementary figure, but if they truly do not have enough trials to check, then they should be honest about it and state it clearly. For example: "We tracked eye position and running, but did not have sufficient trial repeats to analyze their effect on visual responses".

5) Line 610: "reliabilities" should be "reliability".

6) Line 661: There is a typo here, I believe it should read “The cells had varied firing rates...”

Reviewer #3 (Remarks to the Author):

This is a successful revision of an important manuscript. I have no further comments.

Responses to reviewer's comments

Reviewer #1 (Remarks to the Author):

As stated in my original review, I think this is an excellent paper that is bound to become a go-to reference for the field. Most of my earlier comments were related to clarifications of data presentation and methods and have been addressed.

The spatial maps of tuning properties in Figure 3 are a great addition. However, I do not think the presented data are sufficient to support the claims in lines 183-185: "Examining the distribution of average spatial and temporal tuning across the cortical surface, we observed sharp boundaries near the retinotopic borders, with biases that exceed the gradients observed within visual areas". Given the presented data, most features appear to vary smoothly across the cortical surface with the possible exception of discontinuity in TF tuning between areas PM and AM. It would be great to also include a spatial map of ISO/ANISO selectivity.

A: (1) We agree the statement about sharp functional boundaries requires more evidence and now removed this statement in the revision.

(2) A new spatial map of ISO/ANSO selectivity is now provided in Supplementary Figure 9a.

In the discussion, the sentence on lines 397-400 ends in a colon and it is unclear what the "three visual information streams" refers to. I am guessing that this is related to the discussion in lines 448-457 suggesting that area PM may be a part of a separate processing stream. Perhaps the discussion text could be restructured to clarify.

A: We edited the discussion for clarity.

Minor:

1. Line 194: "which shows much less difference between areas" – needs to be rephrased, perhaps "which shows much fewer differences between areas"?

A: We modified the text to '... showed less pronounced differences ...'.

Reviewer #2 (Remarks to the Author):

The authors have significantly improved the analyses and discussion in the revised manuscript. In my view, they have done a nice job of addressing reviewer concerns and highlighting the scientific advance in their work. I have some (mostly minor) concerns listed below that should be addressed, but am otherwise in support of publication.

1) Line 21: "form a continuum" is an unusual phrasing. I would suggest "lie along a continuum".

A: We modified the text accordingly.

2) Line 80: “propound” should be “profound”.

A: We modified the text accordingly.

3) Line 382: the clause “albeit nonuniformly” should have commas before and after.

A: We modified the text accordingly.

4) Line 509-510: This is a declarative statement that has no data supporting it. It would be nice if the authors included an analysis of pupil dilation, eye movements and running in a supplementary figure, but if they truly do not have enough trials to check, then they should be honest about it and state it clearly. For example: “We tracked eye position and running, but did not have sufficient trial repeats to analyze their effect on visual responses”.

A: The experiment design did not allow to draw any conclusions in that regard. To avoid confusion, we have removed all assertions related to behavioral monitoring.

5) Line 610: “reliabilities” should be “reliability”.

A: We modified the text accordingly.

6) Line 661: There is a typo here, I believe it should read “The cells had varied firing rates...”

A: We modified the text accordingly.

Reviewer #3 (Remarks to the Author):

This is a successful revision of an important manuscript. I have no further comments.

A: Thank you.